# ARID1A safeguards the canalization of the cell fate decision during osteoclastogenesis

Jiahui Du ®[1,2,3,4], Yili Liu[1,2,3,4], Jinrui Sun[1,2,3,4], Enhui Yao[1,2,3,4], Jingyi Xu[1,2,3], Xiaolin Wu[1,2,3], Ling Xu[1,2,3], Mingliang Zhou ®[1,2,3], Guangzheng Yang ®[1,2,3] ✉ & Xinquan Jiang ®[1,2,3] ✉

Chromatin remodeler ARID1A regulates gene transcription by modulating nucleosome positioning and chromatin accessibility. While ARID1A-mediated stage and lineage-restricted gene regulation during cell fate canalization remains unresolved. Using osteoclastogenesis as a model, we show that ARID1A transcriptionally safeguards the osteoclast (OC) fate canalization during proliferation-differentiation switching at single-cell resolution. Notably, ARID1A is indispensable for the transcriptional apparatus condensates formation with coactivator BRD4/lineage-specifying transcription factor (TF) PU.1 at *Nfatc1* super-enhancer during safeguarding the OC fate canalization. Besides, the antagonist function between ARID1A-cBAF and BRD9-ncBAF complex during osteoclastogenesis has been validated with in vitro assay and compound mutant mouse model. Furthermore, the antagonistic function of ARID1A-"accelerator" and BRD9-"brake" both depend on coactivator BRD4-"clutch" during osteoclastogenesis. Overall, these results uncover sophisticated cooperation between chromatin remodeler ARID1A, coactivator, and lineage-specifying TF at super-enhancer of lineage master TF in a condensate manner, and antagonist between distinct BAF complexes in the proper and balanced cell fate canalization.

Cell fate canalization, the successive process of being endowed with a specific cellular identity and function, is a core biological event in development and disease[1,2]. Cellular identity is determined by the temporal and spatial gene expression profiles, which are largely orchestrated by transcription factors (TFs), cis-regulatory elements (promoters and enhancers), as well as epigenetic modifications[3]. Chromatin remodelers play a major role in establishing and maintaining permissive gene transcription states. Multiple families of chromatin remodelers exist, most notably BAF, ISWI, CHD, and INO80 family, all of which are ATP-dependent chromatin remodeling complexes that utilize energy derived from ATP hydrolysis to regulate

chromatin architecture and facilitate gene expression via mobilizing and repositioning nucleosomes[4].

The mammalian BAF family is characterized by a highly polymorphic structure, with a variety of common as well as complex-specific subunits assembling into three distinct BAF complexes, termed canonical BAF (cBAF), polybromo-associated BAF (PBAF), and noncanonical BAF (ncBAF)[4]. ARID1A, which is the core component of the cBAF complex, contains a DNA-binding domain and mediates the chromatin remodeling function[5]. Recent studies have found that ARID1A plays a critical role during cell fate determination of multiple tissues in development and disease[6]. For example, ARID1A regulates

[1]Department of Prosthodontics, Shanghai Ninth People's Hospital, Shanghai Jiao Tong University School of Medicine, Shanghai, China. [2]College of Stomatology, Shanghai Jiao Tong University, Shanghai, China. [3]National Center for Stomatology, National Clinical Research Center for Oral Diseases, Shanghai Key Laboratory of Stomatology, Shanghai Research Institute of Stomatology, Shanghai Engineering Research Center of Advanced Dental Technology and Materials, Shanghai, China. [4]These authors contributed equally: Jiahui Du, Yili Liu, Jinrui Sun, Enhui Yao. ✉e-mail: ygzwork@163.com; xinquanjiang@aliyun.com

the fate commitment and differentiation of embryonic stem cells[7], hematopoietic stem cells[8,9], dental mesenchymal stem/progenitor cell[10,11] by modulating nucleosome occupancy, and the poised chromatin configuration at specific loci. Besides, ARID1A has also been shown to have roles to play in thermogenesis accompanied by the enhancer-promoter (E-P) proximity of thermogenic genes depending on beta-adrenergic signaling[12]. While most previous studies are based on bulk-level results, the inevitable cellular heterogeneity could blur the vision of the cellular and molecular mechanism mediated by ARID1A in the cell fate canalization of stem and progenitor cells.

In addition, it is still unclear how chromatin remodeler ARID1A, expressed in a broad range of cells, could regulate specific cell type identity in a particular development process. Mammalian cell identity is controlled by super-enhancers (SEs), which are large clusters of transcriptional enhancers that are occupied by an unusually high density of interacting factors and drive higher levels of transcription than most typical enhancers[13,14]. A previous study has shown that at SE regions, BRD4 and mediator form liquid-like condensates that compartmentalize and concentrate the transcription apparatus to maintain the expression of key cell-identity genes[15]. However, the link between the function of ARID1A and the specific SE activity in cell fate decisions is unclear.

Besides bona fide roles of the cBAF complex during cell fate commitment in most case[7–11,16], our recent study has shown that BRD9-ncBAF negatively regulates cell differentiation during osteoclastogenesis through STAT1/IFN-β signaling[17]. The complexity of composition and function of distinct BAF complexes is the adapting outcome of the evolutionary diversification and biological complexity of the organisms, particularly in mammalian species. While the stage and context-specific targeting on chromatin and function of distinctive BAF complex in a defined cell fate canalization remain largely undefined.

Osteoclasts (OCs), as the exclusive bone-resorbing cells, cooperate with bone-forming osteoblasts maintaining bone development and homeostasis. In the last decades, it has been quite well-documented that OCs are hematopoietic myeloid lineage and derived from bone marrow cells (BMCs) followed by a series of activation, proliferation, and maturation processes under recombinant receptor activator of nuclear factor κB (NF-κB) ligand (RANKL) and macrophage colony-stimulating factor (M-CSF) induction[18]. Although fundamental effort has been made to understand the master regulators during osteoclastogenesis, the dynamic regulation of chromatin remodelers on gene transcription and differentiation trajectory at single-cell resolution remains unclear[19].

Here, we have used osteoclastogenesis as a prototypical model to investigate the role of ARID1A in the cell fate canalization of stem and progenitor cells at single-cell resolution. We have found that *Arid1a* deficiency in myeloid progenitor compromises OC lineage commitment during proliferation-differentiation switching, leading to excessive bone mass. Notably, our data suggest that ARID1A is indispensable for the transcriptional apparatus condensates formation with coactivator BRD4/lineage-specifying TF PU.1 at master TF *Nfatc1* SE region, thereby safeguarding OC fate canalization. Besides, the antagonist between ARID1A-cBAF and BRD9-ncBAF complex during OC fate canalization has been validated. Furthermore, we show that the antagonistic action of ARID1A-"accelerator" and BRD9-"brake" both depend on coactivator BRD4-"clutch" during OC fate canulization. Overall, these findings further expand our knowledge of sophisticated cooperation between chromatin remodeler ARID1A, coactivator, and lineage-specifying TF at master TF SEs in a condensate manner, and the antagonist between distinct BAF complexes in the proper and balanced cell fate canalization.

## Results
### Loss of *Arid1a* leads to excessive bone mass with compromised osteoclastogenesis
Myeloid-specific *Cre*-recombinase (*Cre*)-expressing mouse line, those expressing Cre driven by the lysozyme M promoter (*LysM-Cre* mice) were crossed with Tdtomato (tdT) reporter mice to trace the myeloid cell fate. We found that ARID1A is broadly expressed in BMCs, including cathepsin K (CTSK)+ OCs from LysM+ myeloid lineage (Tdt +CTSK+ cell) on the surface of the trabecular bone of the distal femur from the 4-week-old mouse (Fig. 1a). Moreover, BMCs were cultured with M-CSF and RANKL to further evaluate the expression pattern and level of ARID1A during OC differentiation (Fig. 1b). Colocalization of ARID1A immunofluorescence and tartrate-resistant acid phosphatase (TRAP) staining showed that ARID1A was expressed in TRAP+ multi-nuclear OC in vitro (Fig. 1c). In addition, the expression of ARID1A is gradually upregulated after RANKL treatment both at mRNA and protein level during the osteoclastogenesis (Fig. 1d, e).

To further determine whether ARID1A plays a crucial role during osteoclastogenesis, mice bearing *loxP* sites encompassing the *Arid1a exon5-exon6* (*Arid1a^{fl/fl}* mice) were crossed with *LysM-Cre* mice. Mice lacking *Arid1a* in *LysM*-Cre-expressing cells, hereafter referred to as *LysM-Cre;Arid1a^{fl/fl}* mice (Supplementary Fig. 1a). It showed that ARID1A was knocked out efficiently at protein levels in BMCs from *LysM-Cre;Arid1a^{fl/fl}* mice (Supplementary Fig. 1b, c). In μCT analysis, *LysM-Cre;Arid1a^{fl/fl}* mice exhibited increased femoral trabecular bone mass compared to control male mice at 3 months of age, with significantly increased bone volume/tissue volume ratio (BV/TV), bone mineral density (BMD), trabecular number (Tb.N), trabecular thickness (Tb.Th) and decreased trabecular separation (Tb.Sp) and porosity percent (Po) of the distal femoral trabecular bone (Fig. 1f, g). While there is no significant change in the cortical bone parameters between control and ARID1A KO mice based on the μCT analysis, such as mean total crosssectional bone area (B. Ar) and mean total crosssectional bone perimeter (B. Pm) of cortical bone in the femoral midshaft (Fig. 1f, g). We speculated that there could be some other redundant regulatory mechanism for osteoclastogenesis after loss of *Arid1a* in cortical bone. Consistently, it has been reported that different properties of bone remodeling or modeling activity exist between trabecular and cortical bone[20]. Histologically, the H&E staining (Fig. 1h) and Von Kossa staining (Fig. 1i) results showed the increased bone mass and mineralization in the distal femurs and mesial tibias after *Arid1a* loss in myeloid cell lineage compared with littermate control mice at 3 and 6 weeks of age. The bone phenotype is further validated to be maintained in female mice at 6 months of age in skeletally mature animals with μCT analysis (Supplementary Fig. 2a, b) and H&E staining (Supplementary Fig. 2c, d).

We further individually examined osteoclastic bone resorption and osteoblastic bone formation. We found that the osteoclastic bone resorption surface decreased on the distal femur bone surface from *LysM-Cre;Arid1a^{fl/fl}* mice at 3 weeks of age (Fig. 2a, b). While the osteoblast surface per bone surface (Fig. 2c, d) and osteoid surface per bone surface (OS/BS) (Fig. 2e, f) on the trabecular bone surface underneath the growth plate were comparable between *LysM-Cre;Arid1a^{fl/fl}* mice and littermate control mice using toluidine blue staining and GRB staining[21] at 3 weeks of age. Safranin O/fast green staining and quantification revealed the large persistence of unresorbed calcified cartilage underneath the growth plate of the distal femurs from *LysM-Cre;Arid1a^{fl/fl}* mice at 3 weeks of age (Fig. 2g, h). This result further suggests that osteopetrotic phenotypes caused by ARID1A inactivation are typically associated with the persistence of unresorbed calcified cartilage. Thus, considering the above findings, it is conceivable that the defective osteoclastogenesis following *Arid1a* ablation compromises bone resorption, leading to excessive bone mass. The measurements of serum biomarkers of bone resorption (TRAP and CTX-I) and bone formation (OCN and P1NP) showed that the bone resorption was suppressed without apparent change on the bone formation ability in control and *LysM-Cre;Arid1a^{fl/fl}* male mice at 3-month-old (Supplementary Fig. 2e). The defective osteoclastogenesis is also validated to be maintained in the distal femur of ARID1A depleted female mice at 6 months of age using immunofluorescence staining and quantification of CTSK (Supplementary Fig. 2f, g).

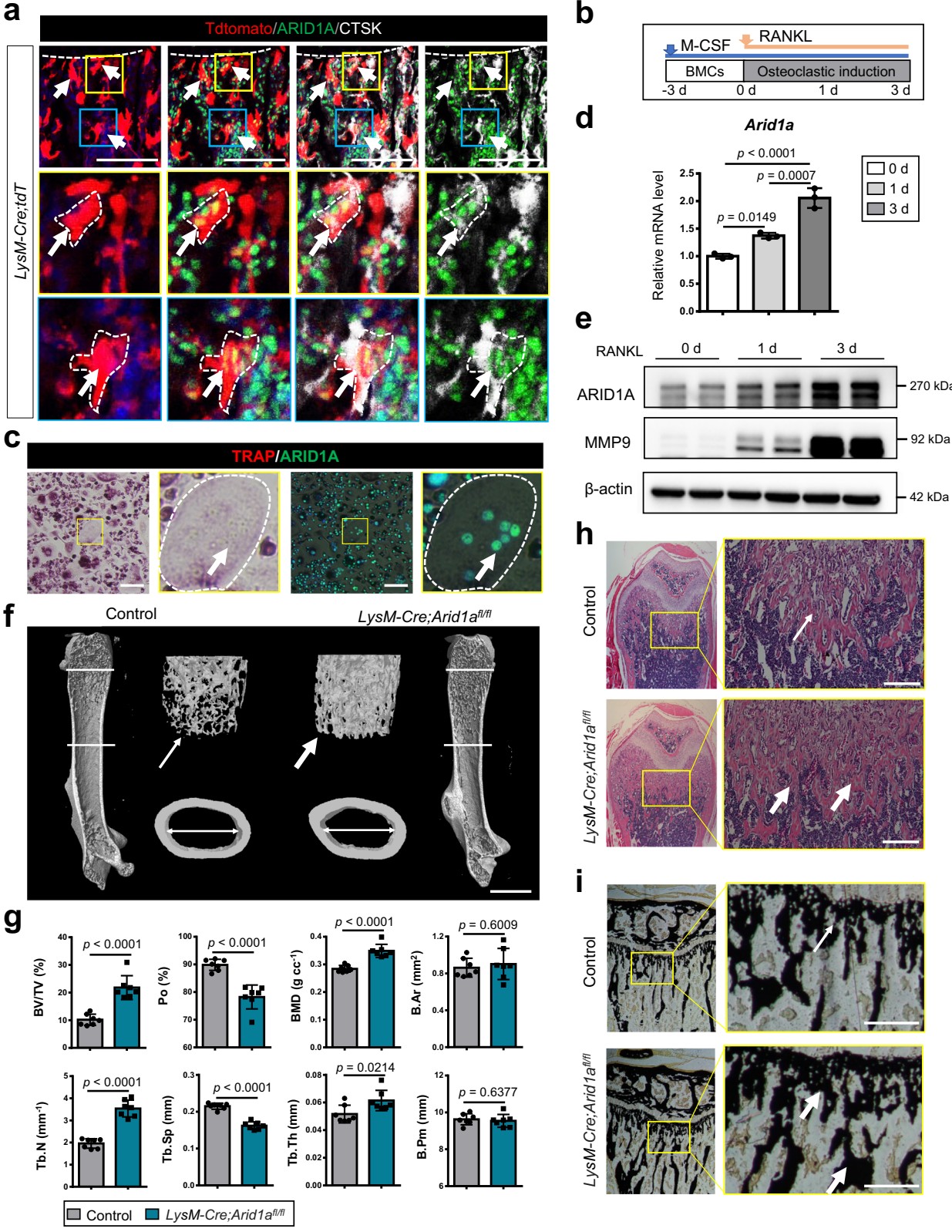

Furthermore, the BMCs from *LysM-Cre;Arid1a^{fl/f}* and control mice were administered with RANKL for osteoclastic induction in vitro. In line with the role of *Arid1a* in vivo, we found that ARID1A loss leads to a decreased number and size of TRAP+ multinuclear cells after RANKL-induction (Fig. 2i). And the expression of osteoclastic-specific genes of CTSK, matrix metallopeptidase 9 (MMP9), tartrate-resistant acid phosphatase 5 (ACP5), dendritic cell-specific transmembrane protein

(DCSTAMP) and OC stimulatory transmembrane protein (OCSTAMP), were decreased after loss of *Arid1a* (Fig. 2j, k).

### ARID1A transcriptionally safeguards the OC fate canalization during proliferation-differentiation switching

To investigate how chromatin remodeler ARID1A transcriptionally promotes the OC lineage determination from myeloid progenitor

**Fig. 1 | Loss of *Arid1a* in myeloid lineage leads to excessive bone mass.**
**a** Immunofluorescence staining of ARID1A and CTSK in the distal femur of 4-week-old *LysM-Cre;tdT* mouse. Arrows indicate LysM+ myeloid lineage ARID1A + CTSK+ osteoclasts (OCs). Scale bar, 100 μm. **b** Schema for osteoclastic differentiation stages of BMCs in vitro. Bone marrow cells, BMCs. **c** Immunofluorescence staining of ARID1A and TRAP staining in BMCs after 3 days of RANKL-induction. Yellow boxes are shown at higher magnification on the right. The dashed line outlines the cell boundaries of OCs. Scale bar, 100 μm. **d** *Arid1a* mRNA expression in BMCs at indicated days after RANKL-induction, as measured by qPCR. *n* = 3 biologically independent samples. **e** ARID1A and MMP9 protein expression in BMCs at indicated days after RANKL-induction, as measured by western blot. **f** Representative micro-CT image of the distal trabecular bone and cortical bone of femurs from 3-month-old male *LysM-Cre;Arid1a^{fl/fl}* mice and littermate control mice. Thicker arrows indicate the increased bone mass. Scale bar, 2 mm. **g** Quantification analysis of bone volume/tissue volume ratio (BV/TV), porosity percent (Po), bone mineral density

(BMD), trabecular number (Tb.N), trabecular separation (Tb.Sp) and trabecular thickness (Tb.Th) of the distal femoral trabecular bone, mean total crossectional bone area (B. Ar) and mean total crossectional bone perimeter (B. Pm) of cortical bone in the femoral midshaft from 3-month-old male *LysM-Cre;Arid1a^{fl/fl}* mice and littermate control male mice. *n* = 7. **h** H&E staining of femurs from 3-week-old male *LysM-Cre;Arid1a^{fl/fl}* mice and littermate control mice. Thicker arrows indicate the increased bone mass. Scale bar, 200 μm. **i** Von Kossa staining of mesial tibias from 6-week-old male *LysM-Cre;Arid1a^{fl/fl}* mice and littermate control mice. The box in the left panel is shown at a higher magnification on the right. Thicker arrows indicate the increased bone mass. Scale bar, 200 μm. All data in this figure are represented as mean ± SD. One-way analysis of variance (ANOVA) with Tukey's multiple comparisons test for **d**. Two-tailed Student's *t*-test for **g**. All experiments were performed in triplicates unless otherwise stated. Source data and exact *p* values are provided in the Source data file.

cells, we assessed the change of mRNA expression after *Arid1a* depletion at an early stage of 24 h post-RANKL induction using high-throughput mRNA sequencing at single cell level (scRNA-Seq) (Fig. 3a).

We analyzed the scRNA-seq profiles of 11,118 cells that passed quality-control criteria, comprising 5813 cells in the control group and 5305 cells in the *LysM-Cre;Arid1a^{fl/fl}* group, respectively. An unsupervised graph-based clustering approach has defined 11 clusters (Fig. 3b). Based on the reference[22], cluster 3 exhibited high expression of OC marker genes (*Ctsk*, *Mmp9*, *Acp5*, and *Dcstamp*), indicating that cluster 3 comprised OC precursors. Cluster 1, 8, 9, 5, 6, 7, and 0 exhibited marker genes of monocytic cells (*Lgmn*, *Dab2*, *Csf1r*, and *BC005537*) at diverse levels, suggesting that monocytic cells contain highly heterogeneous populations of cells. Cluster 2 and 4 represent proliferating populations, exhibiting high marker genes of *Mki67*, *Top2a*, *Aurka*, and *Cdk1*. In particular, cluster 2 represents a monocytic precursor population with high proliferating properties, while cluster 4 represents a cycling population with expression initiation of OC marker genes. Cluster 10 exhibited marker genes for neutrophils (*S100a8*, *S100a9*, *Camp*, and *Ngp*) (Fig. 3c).

To decipher the accurate regulation stages mediated by chromatin remodeler ARID1A during the cell fate decision process of osteoclastogenesis, changes in the proportion of each cell cluster between the control group and *LysM-Cre;Arid1a^{fl/fl}* group were compared. We found that major changes appear in cluster 5, 0, 2, 4, and 3 with increased proportion in cluster 5, 0, and 2 while decreased proportion in clusters 4 and 3 (Fig. 3d and Supplementary Fig. 3a). The sharply decreased number of post-mitotic OC precursors in cluster 3 is consistent with the defective osteoclastogenesis phenotype in above findings (Fig. 1 and Fig. 2). To illustrate the specific stage initiating the defective differentiation process, we further performed pseudotime analysis using monocle2, a method which can predict the differentiation trajectory based on individual cell's asynchronous progression of the process. Pseudotime analysis showed monocytic progenitor differentiates into OC precursor via stepwise pathways (Fig. 3e). During this trajectory, the expression levels of monocytic progenitor (*Lgmn*, *Csf1r* and *Ccl2*) were progressively downregulated, the expression levels of osteoclastic genes (*Ctsk*, *Mmp9* and *Acp5*) were progressively elevated, with expression levels of proliferating marker gene peak at the middle stage (Fig. 3f).

When comparing the trajectory between the control group and *LysM-Cre;Arid1a^{fl/fl}* group, the switching from proliferation to differentiation is hindered after the loss of *Arid1a* (Fig. 3g). Using Tdt as a reporter to trace the myeloid cell fate and EdU to label cell in DNA synthesis phase, we found that the cell number of EdU+CTSK+ OCs from *LysM*+ myeloid lineage (Tdt+EdU+CTSK+ cell, refer as the cluster 4) and EdU-CTSK+ OCs from *LysM*+ myeloid lineage (Tdt+ EdU-CTSK+ cell, refer as the cluster 3) were decreased apparently on the bone surface of the femurs in the mutant mice compared to control mice at 4 weeks of age, suggesting compromised OC lineage commitment at

post-mitotic stage after loss of *Arid1a* (Fig. 3h). To further validate the bioinformatic results, we trace the proliferating cell labeled by EdU in the DNA synthesis phase during OC differentiation in vitro. Twelve hours later, CTSK+ or TRAP+ cells with EdU-labeling, which represent they go through the normal transition of proliferation-differentiation during the detection period, are scarce after loss of *Arid1a* (Fig. 3i).

Therefore, we can conclude that ARID1A is indispensable for safeguarding the canalization during the transition of proliferation-differentiation, and ARID1A loss leads to defective cell fate commitment during osteoclastogenesis.

## ARID1A activates OC master TF *Nfatc1* transcription at SE during the proliferation-differentiation transition

To further investigate the transcriptional regulation mechanism mediated by ARID1A, we focused on the change in the transcriptional profile in the differentiation trajectory after the loss of *Arid1a*. KEGG enrichment analysis showed lysosome, chemokine signaling pathway, and metabolic pathways are highly enriched in cluster 5 and 0, cell cycle and DNA replication are enriched in cluster 2, ribosome-associated pathways are highly enriched in cluster 3, and both DNA replication and ribosome-associated pathways are enriched in cluster 4 (Supplementary Fig. 3b–f). Both pseudotime analysis (Fig. 4a) and the partition-based graph abstraction (PAGA) (Fig. 4b) showed that cluster 4, the key cell population of proliferation-differentiation switching is the initiative defective cell cluster after loss of *Arid1a*. Then the changes in the transcriptional profile of cluster 4 after loss of *Arid1a* were analyzed. There are 1266 upregulated genes and 503 downregulated genes after loss of *Arid1a*, which highly enriched on OC differentiation as expected (Fig. 4c).

It is known that many chromatin regulators are expressed broadly in a variety of cell types and contribute generally to gene expression[23]. Thus, it is unclear how chromatin remodeler ARID1A could regulate specific cell type identity in a particular development process. SEs, which are characterized by disproportionately high levels of histone modification marker H3K27Ac, are large clusters of transcriptional enhancers that drive the expression of genes that define mammalian cell identity[13,14]. To investigate whether ARID1A mediates OC canalization safeguarding through modulation of the activity of SE at cell identity genes, we obtained the genome-wide binding profile of ARID1A and the SEs-associated histone modification marker H3K27Ac in BMCs during osteoclastogenesis after RANKL treatment using chromatin immunoprecipitation followed by sequencing (ChIP-seq). We have identified 15039 enhancer regions with 783 SE regions that differed from typical enhancers in both size and H3K27Ac levels (Fig. 4d). Enhancers tend to loop to and associate with adjacent genes to activate their transcription, with most of these interactions occurring within a distance of 50 kb[24,25]. Using a simple proximity rule, we assigned all transcriptionally active genes (TSSs) to SEs within a 50 kb window and identified 728 genes associated with SEs (Supplementary

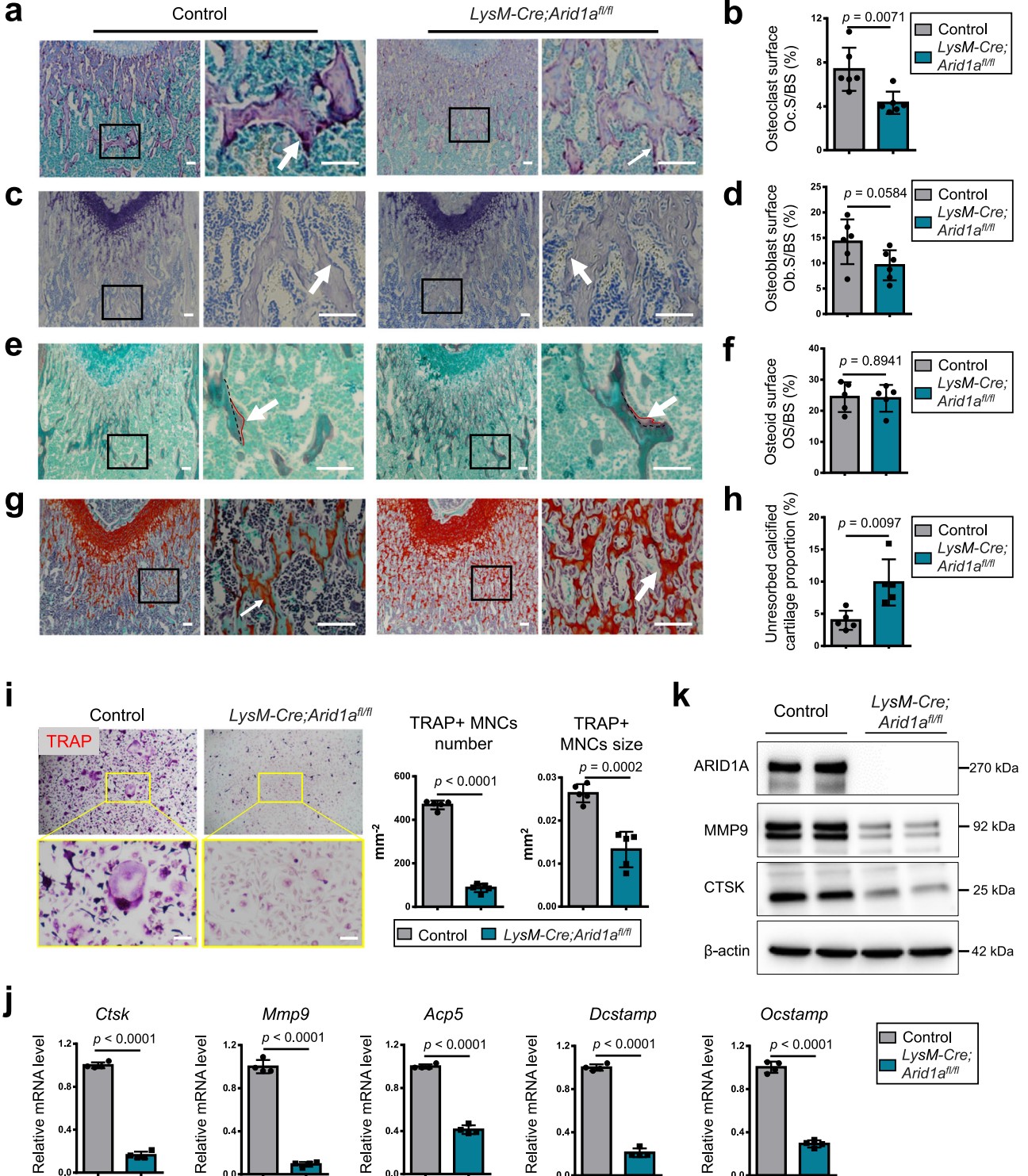

**Fig. 2 | Loss of *Arid1a* in myeloid lineage leads to compromised OC lineage commitment. a** TRAP staining (red) and **b** osteoclastic surface quantification, **c** toluidine blue staining and **d** osteoblastic surface quantification, **e** GRB staining and **f** osteoid surface quantification, **g** safranin O/fast green staining and **h** quantification of persistence of unresorbed calcified cartilage of distal femoral trabecular bone from 3-week-old male *LysM-Cre;Arid1a^{fl/fl}* mice and littermate control mice. The box on the left is shown at a higher magnification on the right. Scale bar, 100 μm. *n* = 6 biologically independent samples in **b** and **d**; *n* = 5 biologically independent samples in **f** and **h**. **i** TRAP staining and quantification analysis of BMCs from 4-week-old male *LysM-Cre;Arid1a^{fl/fl}* mice and littermate control mice after RANKL-induction. Scale bar, 100 μm. *n* = 5 biologically independent samples.

**j** The mRNA expression of osteoclastic-specific genes of *Ctsk, Mmp9, Acp5, Dcstamp*, and *Ocstamp* in BMCs from 4-week-old male *LysM-Cre;Arid1a^{fl/fl}* mice compared with that from control littermates after RANKL-induction, as measured by qPCR. *n* = 4 biologically independent samples. **k** The protein expression of ARID1A, MMP9, and CTSK in BMCs from 4-week-old male *LysM-Cre;Arid1a^{fl/fl}* mice compared with that from control littermates after RANKL-induction, as measured by western blot. All data in this figure are represented as mean ± SD. Two-tailed Student's t-test for **b, d, f, h, i** and **j**. All experiments were performed in triplicates unless otherwise stated. Source data and exact *p* values are provided in the Source data file.

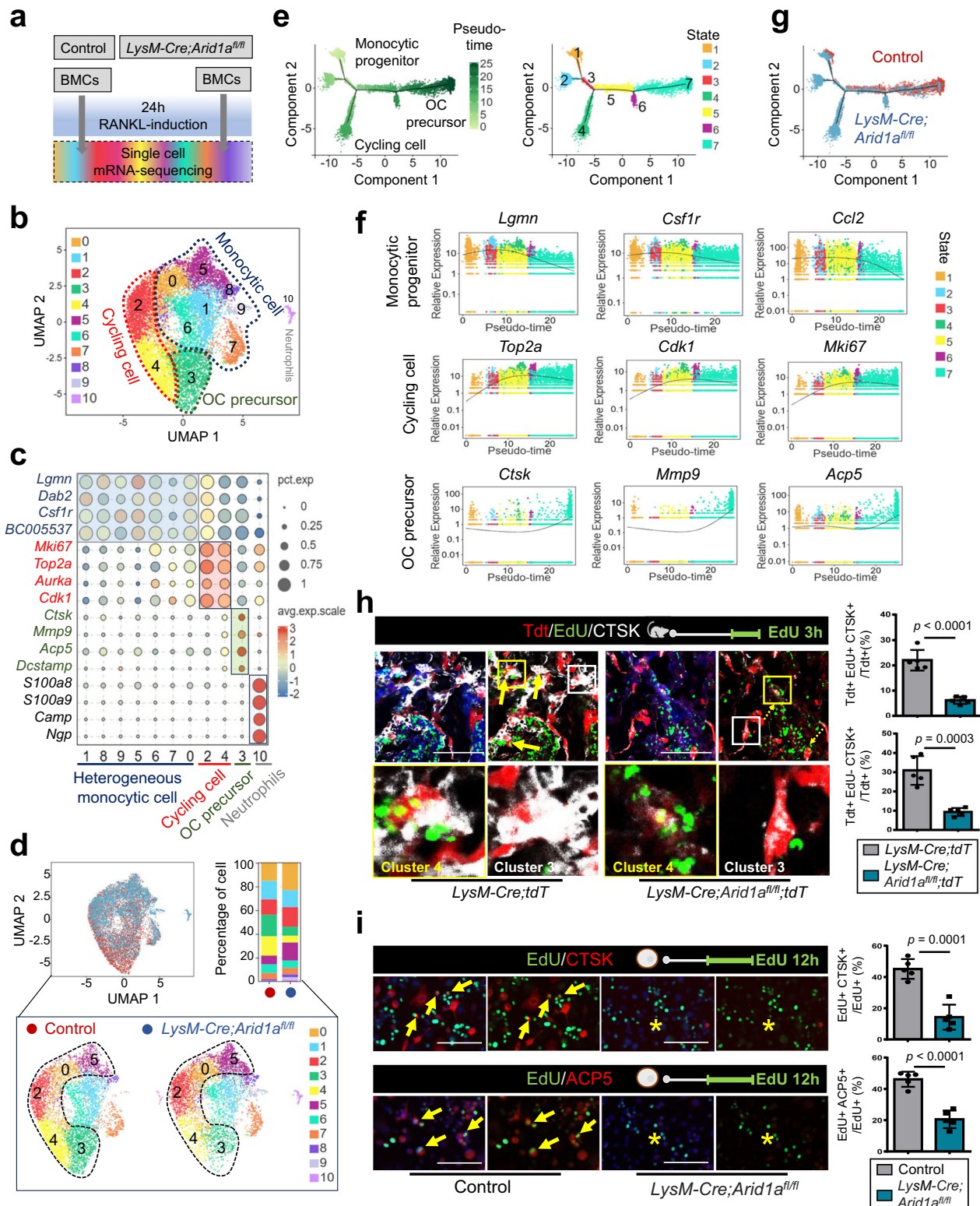

Data 1). The SE-associated genes include multiple genes that have previously been shown to be involved in osteoclastogenic programs[26], including *Nrp2*, *Sema4d*, *RNF19b*, *Nfatc1*, and *Fos* (Fig. 4d). Then, we compared the SE-associated 728 genes and ARID1A occupied 758 genes and identified 143 genes in common (Fig. 4e and Supplementary Data 2).

Linking the ARID1A occupied, SE-associated genes profile with the change in transcription profile after ARID1A loss allows inference of the potential functional downstream targets of ARID1A. Therefore, we

compared the 143 ARID1A-occupied, SE-associated genes with the 44 OC differentiation-associated differentiated expressed genes after loss of *Arid1a* within cluster 4, the key population determining OC fate (Fig. 4c and Supplementary Data 3), which suggests the potential key targets of ARID1A during OC fate determination. *Nfatc1*, encoding OC master TF, the TF necessary and sufficient for cell type specification and differentiation[27,28], is the only enriched gene that down regulated after loss of *Arid1a* (Fig. 4e). Previous studies have shown that ectopic expression of NFATc1 causes precursor cells to undergo efficient

**Fig. 3 | ARID1A transcriptionally safeguards the OC fate canalization during proliferation-differentiation switching. a** Schematic workflow of single-cell mRNA-sequencing experiments. **b** Uniform manifold approximation and projection (UMAP) visualization of 11 clusters in the OC culture system, representing four major different cell types based on known differentiation marker genes expression, as shown in the dot plot **c. d** UMAP plot comparison and change of the proportion of each cell cluster between the control group and *LysM-Cre;Arid1a^fl/fl^* group. **e** Pseudotime analysis showing the OC differentiation process passing through stepwise states. Left: Pseudotime trajectory colored by timeline; Right: pseudotime trajectory colored by cell states. **f** The dynamic expression patterns of signature genes during the trajectory of osteoclastogenesis by cell states. **g** Pseudotime trajectories of the control group and *LysM-Cre;Arid1a^fl/fl^* group. **h** Immunofluorescence and quantification of CTSK (white) and visualization of EdU (green) and tdT (red) in the distal femur of 4-week-old male *LysM-Cre;tdT* mice *and*

*LysM-Cre;Arid1a^fl/fl^;tdT* mice. The Schematic at the top right indicates the EdU labeling protocol. The colored boxes in the first panel are enlarged below. Yellow boxes and arrows indicate Tdt+EdU+CTSK+ cells, referred to as cluster 4, and white boxes indicate Tdt+EdU-CTSK+ cells, referred to as cluster 3. Scale bar, 100 μm. *n* = 5 biologically independent samples. **i** Immunofluorescence and quantification of CTSK (red) or ACP5 (red) and visualization of EdU (green) in BMCs from 4-week-old male *LysM-Cre;Arid1a^fl/fl^* mice and littermate control mice after RANKL-induction. The Schematic at the top right indicates the EdU labeling protocol. Yellow arrows indicate EdU+CTSK+ or EdU+ACP5+ cells. The asterisk indicates no signal. Scale bar, 100 μm. *n* = 5 biologically independent samples. All data in this figure are represented as mean ± SD. Two-tailed Student's *t*-test for **h** and **i**. All experiments were performed in triplicates unless otherwise stated. Source data and exact *p* values are provided in the Source data file.

differentiation without RANKL signaling and suggested that NFATc1 may represent a master switch for regulating the terminal differentiation of OCs[27,28]. We find *Nfatc1* expression was upregulated during the differentiation trajectory, while failed to be upregulated in cluster 4 after loss of *Arid1a*, suggesting it may be the core downstream target of ARID1A during the OC fate determination (Fig. 4f). We traced the proliferating cell labeled by EdU for 12 hours, and found that EdU-labeled NFATc1+ cells are detectable in the control group while scarce after loss of *Arid1a* (Fig. 4g). We further validated the role of NFATc1 in the defective OC differentiation after loss of *Arid1a* through constituently expressing *Nfatc1* in BMCs from the mutant mouse using lentivirus vector. The constituently expressed *Nfatc1* partially rescued the defective OC differentiation after ARID1A depletion with increased MMP9, CTSK, and ACP5 expression (Fig. 4h and Supplementary Fig. 4).

To further confirm if these enhancers on the *Nfatc1* gene are functional, the promoter luciferase activity assay was performed after generating mini genes with the E-P connection. Based on our ChIP-seq data of ARID1A and H3K27Ac in BMCs during osteoclastogenesis (Fig. 4i), we annotated their binding peaks as presentative enhancer regions at upstream (Enhancer 1, E1) and downstream (Enhancer 2 and 3, E2 and E3) of *Nfatc1* promoter[29]. The results showed that E1, E2, and E3 together exhibit a much stronger activation function for *Nfatc1* promoter activity (Fig. 4i). Recent studies have shown that BAF complex-mediated higher-order chromatin interactions play significant roles in gene transcription, and ARID1A has been shown previously to have roles to play in thermogenesis accompanied by the E-P proximity of thermogenic genes depending on beta-adrenergic signaling[12]. The remote enhancers E2 and E3 marked by H3K27ac located 10 and 21 kb downstream of the *Nfatc1* promoter (Fig. 4i), which are ideal distances to examine the long-range looping model by chromosome conformation capture (3 C) analysis[30]. Therefore 3C-qPCR was performed to evaluate the E-P proximity on the *Nfatc1* gene during osteoclastogenesis. Using the *Nfatc1* promoter as an anchor, we identified there is loop formation between E2, E3 with the *Nfatc1* promoter region during osteoclastogenesis (Fig. 4j). While ARID1A depletion could reduce the long-range looping signal in response to RANKL-induction (Fig. 4k), suggesting the ARID1A-mediated transcriptional activation of *Nfatc1* could be partially through facilitating RANKL-induced E-P interactions.

Previous OC scRNA-seq studies have also identified that both Cbp/p300-interacting transactivator with Glu/Asp-rich carboxy-terminal domain 2 (Cited2)[22] and guanosine triphosphatase (GTPase) family member RAB38[31], as a NFATc1-dependent molecule, control the transition of proliferation-differentiation in OCs. We found that the loss of *Arid1a* could also lead to decreased expression of *Cited2* and *Rab38* (Supplementary Fig. 5), suggesting besides failed activation of *Nfatc1* validated in our study, the downregulated expression of other critical factors could also contribute to the defective transition of proliferation-differentiation in OCs after the loss of *Arid1a*.

## ARID1A activates *Nfatc1* through constructing condensates with coactivator BRD4 and lineage-specifying TFs at SE

At active genes, enhancers and core promoters are in close proximity, so factors associated with enhancers can regulate transcriptional initiation or elongation largely via acting on the transcription apparatus[23]. BRD4, which links active enhancers with cell identity gene induction in adipogenesis and myogenesis, was reported to act as an enhancer epigenomic reader, interact with mediator, and participate in the control of transcriptional elongation by SE reprogramming and TF rewiring activity[23,32,33].

We have detected the resemblance between the defective OC differentiation phenotype between BRD4 inhibition with JQ1 and ARID1A loss, with both decreased number and size of TRAP+ multinuclear OCs (Fig. 5a), and inhibited expression of *Mmp9*, *Ctsk*, *Acp5*, and *Dcstamp* (Fig. 5b, c). To further investigate the similarity between the regulation mechanism of ARID1A and BRD4 during osteoclastogenesis, both binding profile of ARID1A and BRD4 (before and after RANKL-induction) was compared. The result shows that the OC differentiation process was only enriched on the common-occupied genes of ARID1A and BRD4 (Fig. 5d and Supplementary Fig. 6), suggesting their potential cooperative function during osteoclastogenesis. Given the indispensable function of ARID1A through SE activity during the OC cell fate decision (Fig. 4) and both BRD4 inhibition with JQ1 and ARID1A loss after RANKL induction leads to suppressed *Nfatc1* transcription (Fig. 5e), we speculated that BRD4 could be its critical coactivator in activating *Nfatc1* SE thereby OC differentiation process. Then the colocalization in the nucleus of ARID1A with BRD4 was validated in osteoclastogenesis using double immunofluorescent staining. Previous studies have validated that coactivator condensation at SEs links phase separation and gene control[15], we found the colocalized BRD4-ARID1A forms nuclear puncta (Fig. 5f), suggesting they may cooperatively compartmentalize and concentrate transcription apparatus in the control of key cell identity genes during OC fate determination process. We further found although the expression of BRD4 is affected at neither mRNA nor protein level (Supplementary Fig. 7a, b), the binding of BRD4 at *Nfatc1* SE region is compromised after loss of *Arid1a* and the colocalization of ARID1A with BRD4 at *Nfatc1* SE region is dependent on RANK signaling (Fig. 5g). These results suggest BRD4 cooperates with ARID1A at SE of *Nfatc1* to safeguard the canalization during osteoclastogenesis, and loss of *Arid1a* leads to defective activation function of BRD4 at *Nfatc1*.

To identify the critical lineage-specifying TFs that cooperate with ARID1A at SE of *Nfatc1* to safeguard the canalization during osteoclastogenesis, we conducted the motif analysis of ARID1A and BRD4 binding profiles. ChIP-Seq data showed there are 828 peaks for ARID1A (*p* < 0.001) and 1298 peaks for ARID1A-dependent BRD4 binding (*p* < 0.001). We found the Ets family TF PU.1 (Spi-1) in the top enriched TF motifs in ARID1A binding sites, and also in ARID1A-dependent BRD4 binding sites which lost after ARID1A depletion (Fig. 5h). PU.1 is a lineage-specifying TF in hematopoiesis, that positively regulates many

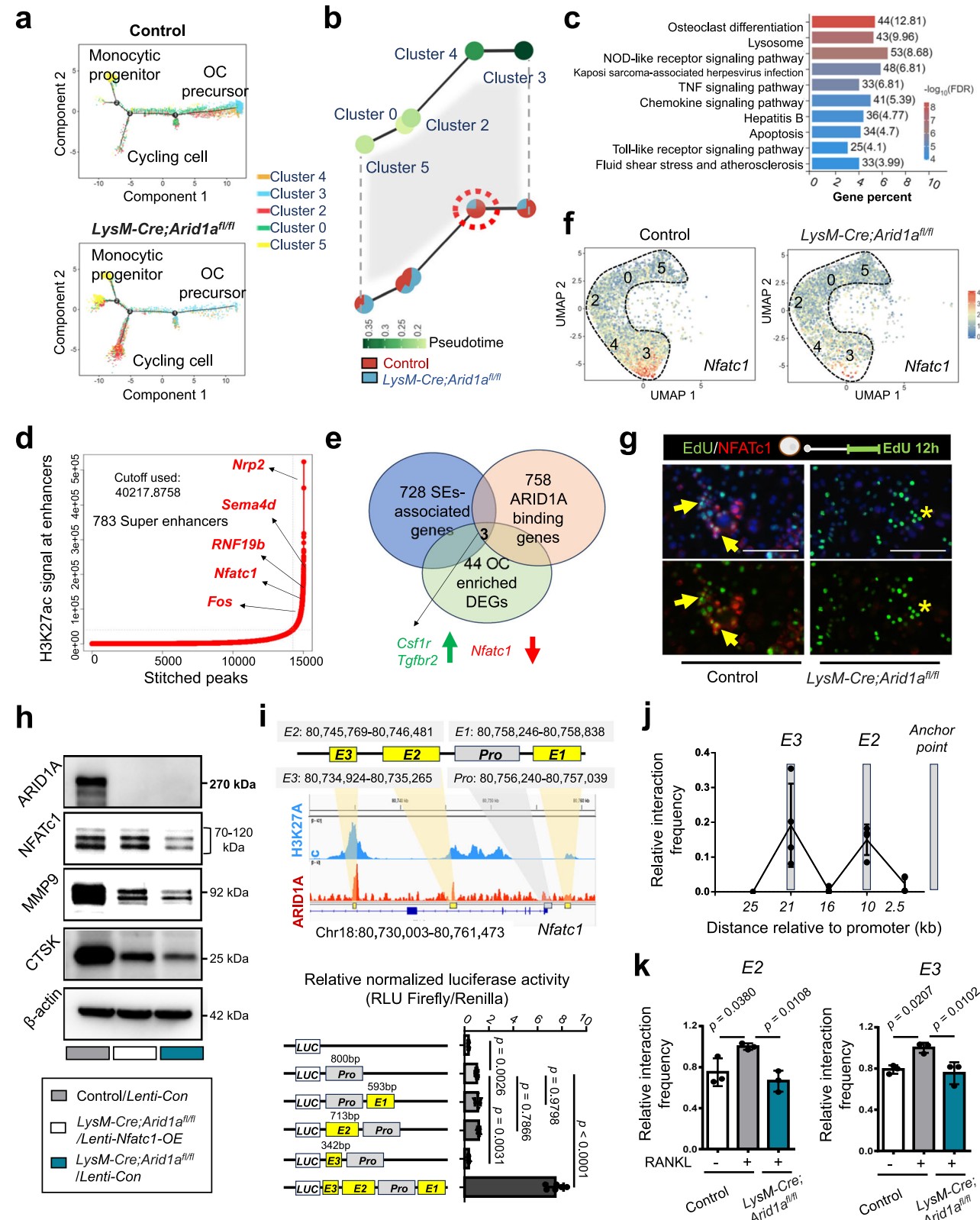

genes in the macrophage, granulocyte, dendritic cell, and B-cell lineages[34,35]. We further conducted PU.1 ChIP-seq and found the binding profile of PU.1 is highly similar to ARID1A and BRD4 in the *NFATc1* SE region (Fig. 5g). Double immunofluorescent staining result showed that PU.1 colocalizes with ARID1A forming nuclear puncta (Fig. 5i), suggesting PU.1, ARID1A and coactivator BRD4 may cooperatively compartmentalize and concentrate transcription apparatus

in the control of key cell identity genes during OC fate determination process. Then the physical interaction between ARID1A with BRD4 or PU.1 was validated in osteoclastogenesis using co-IP assay in BMCs during osteoclastogenesis (Fig. 5j). While the expression level of PU.1 does not apparently change after ARID1A loss at neither mRNA nor protein level (Supplementary Fig. 7c, d), we found that loss of *Arid1a* lead to inhibited binding of PU.1 at SE of *Nfatc1* (Fig. 5k) and decreased

**Fig. 4 | ARID1A activates *Nfatc1* transcription at SE during proliferation-differentiation switching. a** Pseudotime trajectory colored by cell clusters. **b** PAGA graph showing trajectory colored by the timeline in upper and the proportion in the target cell cluster in lower. **c** Bar plot of the top 10 enriched KEGG pathways in cluster 4 between control and *LysM-Cre;Arid1a^{fl/fl}* group. **d** Total H3K27Ac ChIP-seq signal in units of reads per million in enhancer regions for all enhancers in BMCs after RANKL-induction. **e** Venn diagram showing overlapping genes between ARID1A occupied, SE-associated genes with OC-associated DEGs after loss of *Arid1a* within cluster 4. **f** UMAP visualization of *Nfatc1* expression in control and *LysM-Cre;Arid1a^{fl/fl}* group. **g** NFATc1 immunofluorescence and EdU visualization in RANKL-inducted BMCs from control and *LysM-Cre;Arid1a^{fl/fl}* mice. Yellow arrows indicate EdU+ NFATc1+ cells. Scale bar, 100 μm. **h** ARID1A, NFATc1, MMP9, and CTSK protein expressions in *Nfatc1*-overexpressed (OE) and control BMCs from control and *LysM-Cre;Arid1a^{fl/fl}* group after RANKL-induction. **i** Luciferase reporter activity driven by *Nfatc1* promoter and enhancer elements.

Gene tracks of H3K27Ac and ARID1A ChIP-seq occupancy at the super enhancer of *Nfatc1* in RANKL-induced BMCs (top panel). Gray box indicates promoter region (*Pro*). Yellow boxes indicate the presentative enhancer regions upstream (*E1*) and downstream (*E2* and *E3*). Data were normalized to Renilla internal control. *n* = 6 biologically independent samples. **j** 3C-qPCR analysis of interaction frequency percentage of the restriction fragments with the anchor point fixed near *Nfatc1* promoter in 24 h RANKL-induced BMCs. The gray shadows highlight the regions containing *E2* and *E3* enhancer elements and the anchor point. *n* = 4 biologically independent samples. **k** Interaction frequency of the restriction fragments in *E2, E3* with *Nfatc1* promoter in BMCs from control and *LysM-Cre;Arid1a^{fl/fl}* mice with or without 24 h RANKL-induction. *n* = 3 biologically independent samples. Short-range ligation product used for normalizion for **j, k**. All data in this figure are represented as mean ± SD. One-way ANOVA with Dunnett's multiple comparisons test for **i, k**. All experiments were performed in triplicates unless otherwise stated. Source data and exact *p* values are provided in the Source data file.

colocalized PU.1-BRD4 puncta (Fig. 5l), suggesting ARID1A-BRD4-PU.1 condensate is disrupted after ARID1A loss.

Besides PU.1, previous studies have shown that NF-kB[36], AP-1[37], and NFATc1[36] itself could contribute to the *NFATc1* induction by activating cis-regulatory elements. We found NFKB1, AP-1, and NFATc1 motifs are also enriched in the *NFATc1* SE region (Supplementary Fig. 8a, b), suggesting NF-kB, AP-1, and NFATc1 itself could also participate in the ARID1A/BRD4/PU.1 condensate formation at *Nfatc1* SE region during autoamplification of the *NFATc1* gene and safeguarding the canalization of osteoclastogenesis.

## Antagonist between ARID1A-cBAF and BRD9-ncBAF complex during OC fate canalization

The mammalian BAF complex is highly polymorphic, assembling its cell type- or developmental stage-specific subunits in a combinatorial manner[38–41]. A previous study reported that the BRD9-ncBAF complex differs from the ARID1A-containing ESC BAF complex (esBAF), and heightened its distinct function during ESC pluripotent[42]. Our recent study demonstrated that the BRD9-ncBAF complex negatively regulates OC differentiation[17], which is inverse to the function of ARID1A validated in our present study. The distinct phenotypes between the loss of ARID1A and BRD9 enlighten us to further explore whether they exert the role of "accelerator" or "brake" during the OC fate canalization and the potential linkage between their antagonistic function.

We initially investigated the change in the role of ARID1A in the OC lineage determination process after BRD9 loss and found ARID1A protein were higher in BMCs (Fig. 6a), suggesting ARID1A/BRD4/PU.1/NFATc1 axis may contribute to the enhanced OC lineage determination process after BRD9 loss. To test our hypothesis, we partially knockdown *Arid1a* using siRNA or inhibit BRD4 activity with JQ1 in BMCs from *LysM-Cre;BRD9^{fl/fl}* mouse and both strategies could partially rescue the enhanced OC differentiation process with decreased number of TRAP+ multinuclear OCs and inhibited expression of *Mmp9, Acp5, Dcstamp,* and *Ctsk* (Fig. 6b–f). Our recent study demonstrated that the BRD9-ncBAF complex negatively regulates OC differentiation through STAT1/IFN-β signaling activation[17]. We found although the over-activated NFATc1 after BRD9 loss is rescued, the expression of STAT1 is still at a low level in BMCs from *LysM-Cre;BRD9^{fl/fl}* neither with partially knockdown *Arid1a* using siRNA nor BRD4 activity inhibition using JQ1 (Fig. 6g–j), suggesting the rescue mechanism is indirect and not STAT1-dependent. Therefore, independent of STAT1/IFN-β signaling, our new finding shows that BRD9 could also restrain the OC lineage commitment through antagonizing the activation of ARID1A on the cell fate determination process.

Reciprocally, we further detected that ARID1A loss leads to the increased protein level of BRD9 (Fig. 7a). Gene set enrichment analysis (GSEA) indicated that IFN-β signaling was enriched in BMCs from *LysM-Cre;Arid1a^{fl/fl}* mouse comparing with the control group after RANKL induction (Fig. 7b and Supplementary Fig. 9). The

increased *Stat1 and Ifnb1* expression in osteoclastic induced BMCs after ARID1A loss was validated with qPCR (Fig. 7c). As above mentioned, that BRD9/STAT1/IFN-β signaling axis restrain osteoclastogenesis, to further elucidate whether it simultaneously contributes to the compromised OC differentiation process after ARID1A depleted at least partially, we treated the ARID1A-depleted BMCs with BRD9 inhibitor during osteoclastogenesis. The result showed that BRD9 inhibition partially rescued OC differentiation defects after loss of *Arid1a* with increased TRAP+ multinuclear OC number and size (Fig. 7d) and expression level of *Mmp9, Acp5, Dcstamp,* and *Ctsk* (Fig. 7e, f). In vivo model with BRD9 heterozygous knockout in *LysM-Cre;Arid1a^{fl/fl}* mice were further generated to validate this hypothesis. The osteopetrotic phenotype (Fig. 8a–d) and defective OC differentiation (Fig. 8e, f) in *LysM-Cre;Arid1a^{fl/fl}* group compared with control female mice at 3-month old of age was partially rescued in *LysM-Cre;Arid1a^{fl/fl};BRD9^{fl/+}* group with alleviated abnormally upregulated STAT1 after ARID1A loss (Fig. 8g).

ARID1A depletion leads to the increased protein level of BRD9 with no apparent change in mRNA level (Fig. 7a and Supplementary Fig. 10a), suggesting post-transcriptional level regulation of BRD9 might have been changed after loss of ARID1A. Protein ubiquitination, mediating about 80% of protein degradation in eukaryotes, is tightly determined by the delicate balance between specific ubiquitin ligase (E3)-mediated ubiquitination and deubiquitinase (DUB)-mediated deubiquitination[43]. Therefore, we predicted the potential E3 ligases/DUBs for BRD9 using UbiBrowser 2.0 software (Supplementary Fig. 10b). The top 15 predicted ubiquitin ligases/DUBs were integrated and analyzed with the differential expression analysis of the RNA-seq data in BMCs comparing between from control and *LysM-Cre;Arid1a^{fl/fl}* mouse. E3 ubiquitin-protein ligase Murine Double Minute 2 (MDM2), with the top high confidence score, is significantly downregulated after ARID1A loss (Supplementary Fig. 10c, d). We further validated that the degradation of MDM2 with SP141 inhibitor[44] could increase the BRD9 protein level (Supplementary Fig. 10e), suggesting the downregulated MDM2 after loss of *Arid1a* may contribute to the increased BRD9 protein level.

Therefore, besides activating OC master TF *Nfatc1* through constructing condensates with coactivator BRD4/lineage-specifying PU.1 at SE, the function of ARID1A on antagonizing with BRD9/Stat1 axis is also critical for the canalization of the OC fate decision (Fig. 8h, i). Loss of *Arid1a* leads to failed *Nfatc1* upregulation and *Stat1* downregulation during proliferation-differentiation switching (Fig. 8j). In sum, during the canalization of OC lineage commitment, the function of ARID1A-cBAF and BRD9-ncBAF complex antagonizes with each other to control and safeguard the fate determination process.

Similarly with the result after loss of ARID1A, the treatment of BRD4 inhibitor JQ1 suppresses osteoclastogenesis (CTSK expression) (Fig. 9a). On the other hand, BRD4 inhibition leads to downregulated STAT1 expression (Fig. 9a, b) and IFN-β signaling (Fig. 9c), which is

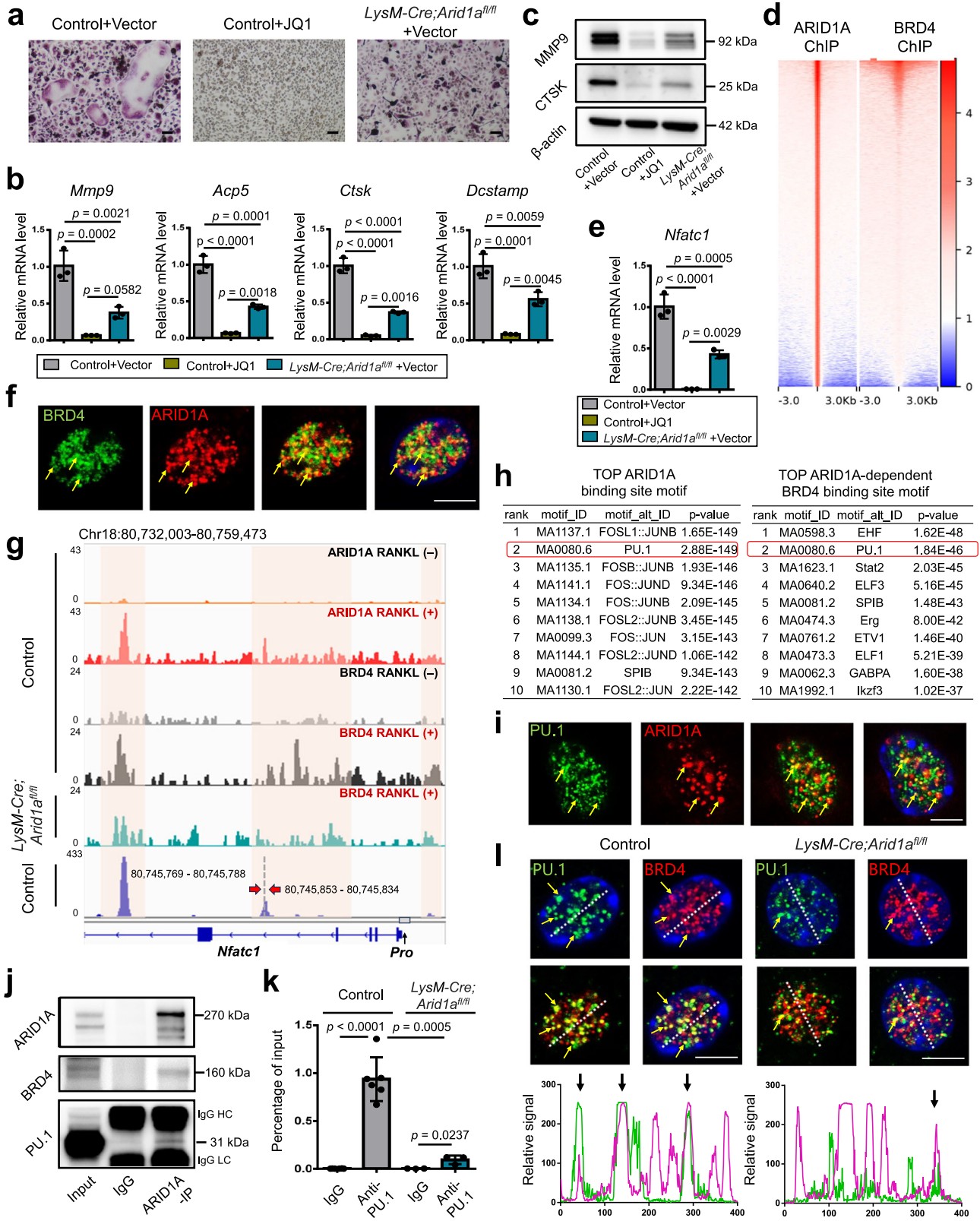

dissimilar with the result after loss of ARID1A (Fig. 7b, c and Supplementary Fig. 9). Previous studies found that BRD9 cooperates with BRD4 in naïve pluripotency maintenance in embryonic stem cells and macrophage activation[42,45]. Here we found that BRD4 physically interacts with BRD9 in osteoclastogenesis (Fig. 9d), and the expression level of STAT1 decreased in ARID1A-depleted BMCs after the treatment of BRD4 inhibitor JQ1 (Fig. 9b, e), which suggests that the enhanced

BRD9-Stat1 axis after ARID1A loss is BRD4 dependent. Through comprising in the distinguished protein complexes, BRD4 activates OC master TF *Nfatc1* through interaction with the ARID1A-cBAF complex and also enhances IFN-β signaling/*Stat1* activity through interaction with the BRD9-ncBAF complex. Our finding indicates that the antagonistic action of ARID1A-"accelerator" and BRD9-"brake" both depend on BRD4, which exerts a "clutch" role (Fig. 9f).

**Fig. 5 | ARID1A activates *Nfatc1* through constructing condensates with coactivator BRD4/lineage-specifying TF PU.1 at SE. a** TRAP staining of JQ1 or vector treated RANKL-induced BMCs from control and *LysM-Cre;Arid1a$^{fl/fl}$* group. Scale bar, 100 μm. **b** *Mmp9, Acp5, Ctsk,* and *Dcstamp* mRNA and **c** MMP9 and CTSK protein expressions in JQ1 or vector treated RANKL-induced BMCs from control and *LysM-Cre;Arid1a$^{fl/fl}$* group. *n* = 3 biologically independent samples. **d** Heatmap representation of ChIP-seq against ARID1A and BRD4 in ± 3 kb around the center of the ARID1A peak. **e** *Nfatc1* mRNA expression in JQ1 or vector treated RANKL-induced BMCs from control and *LysM-Cre;Arid1a$^{fl/fl}$* group. *n* = 3 biologically independent samples. **f** BRD4 and ARID1A immunofluorescence in RANKL-induced BMCs. Yellow arrows indicate BRD4/ARID1A nuclear puncta. Scale bar, 5 μm. **g** Gene tracks of ARID1A, BRD4 and PU.1 ChIP-seq occupancy at *Nfatc1* super enhancer in BMCs from control and *LysM-Cre;Arid1a$^{fl/fl}$* group before or after RANKL-induction. Gray box indicates the promoter region (*Pro*). The orange shadows highlight the regions containing enhancer elements. **h** Top 10 motifs enriched in ARID1A binding sites

and ARID1A-dependent BRD4 binding sites which loss after ARID1A depletion. **i** PU.1 and ARID1A immunofluorescence in BMCs after RANKL-induction. Yellow arrows indicate PU.1/ARID1A nuclear puncta. Scale bar, 5 μm. **j** Co-IP assay with ARID1A antibody (or IgG) in BMCs during osteoclastic induction, followed by immunoblotting of ARID1A, BRD4, and PU.1. **k** ChIP-qPCR with PU.1 antibody (or IgG) in RANKL-induced BMCs from *LysM-Cre;Arid1a$^{fl/fl}$* (*n* = 3) and control mice (*n* = 6). The positions of the sets of primers used for the ChIP–qPCR are denoted in **g**. **l** Immunofluorescence and co-localization analysis of PU.1 and BRD4 in RANKL-induced BMCs from control and *LysM-Cre;Arid1a$^{fl/fl}$* group. Yellow arrows indicate PU.1/BRD4 nuclear puncta. Scale bar, 5 μm. All data in this figure are represented as mean ± SD. One-way ANOVA with Tukey's multiple comparisons test for **b** and **e**. The one-tailed Fisher's Exact test for **h**. Two-tailed Student's *t*-test for **k**. All experiments were performed in triplicates unless otherwise stated. Source data and exact *p* values are provided in the Source data file.

## Discussion

Epigenetic regulation, particularly chromatin remodeling, is essential for linking genotype with phenotype in development and evolution[46,47]. Cell fate canalization comprises lineage decisions and stepwise differentiation with a continuum of stage and lineage-restricted gene expression[1,2]. However, the precise regulation mechanism of dynamic chromatin remodeling at the lineage master regulators remains unclear. Using osteoclastogenesis as a prototypical model, our study suggests that chromatin remodeler ARID1A, although broadly expressed in LysM+ myeloid progeny, is indispensable for the transcriptional apparatus condensates formation with coactivator BRD4/lineage-specifying TF PU.1 at master regulator *Nfatc1* SE region during safeguarding the cell fate canalization. Besides PU.1, we found the NF-kB, AP-1, and NFATc1 motif are also enriched in the identified *Nfatc1* SE region, which suggests they could also facilitate *Nfatc1* transcription activation.

Enhancers tend to loop to and associate with adjacent genes to activate their transcription, with most of these interactions occurring within a distance of 50kb[24]. Therefore, transcriptionally active genes were assigned to enhancers using a simple proximity rule[25]. While gene expression is modulated by the interaction between cis-regulatory sequences with TFs and chromatin regulators within the three-dimensional (3D) chromatin landscape and vertebrate promoters and their regulatory elements can be separated by thousands or millions of base pairs[48,49]. Therefore, new tools for high-throughput super-resolution imaging of chromatin which could directly visualize the 3D chromatin organization will unveil a more comprehensive and accurate regulation landscape[50].

Recent studies have already found that chromatin remodeler ARID1A plays a critical role during the development and disease of multiple tissues[7–11]. While most published studies are based on bulk-level results, inevitable cellular heterogeneity could blur the vision of the cellular and molecular mechanism in the cell fate canalization of stem and progenitor cells. Here, we investigated the transcriptional regulation of ARID1A in OC lineage determination at the single-cell resolution and identified the key subpopulation determining the transition of proliferation to differentiation, which is less than one-sixth of the total cell population. During the differentiation trajectory, we found there is a sharply increased expression of OC master TF *Nfatc1*, the downstream target gene of ARID1A, which is consistent with the change at the bulk-resolution. While as for *Stat1*, which increases at the initiation stage of differentiation, then decreases at the fate determination stage permitting the terminal differentiation process, it is no wonder that, at the bulk-resolution in our previous study[17], its stage-dependent reduction was not identified during osteoclastogenesis. Consistently, Tsukasaki and colleagues claimed that the progressively elevated expression level of *Cited2* during OC differentiation trajectory is difficult to be identified by bulk RNA-seq analysis, probably due

to the relatively low expression levels in OCs and cellular contamination[22]. Therefore, the transcription and trajectory analysis of the cell fate canalization at the single-cell resolution has advantages in revealing unprecedented stage-specific regulation mechanisms at high resolution.

Besides the acquisition of specific cell type identity with the lineage master regulators, such as *Nfatc1*, the differentiation-associated cell cycle arrest is a prerequisite for the transition of proliferation-differentiation in OCs as well. For example, we could identify that the progressively elevated expression levels of osteoclastic genes are accompanied by the downregulated proliferating marker genes during the differentiation trajectory (Fig. 3f). Tsukasaki and colleagues reported that CITED2 may facilitate osteoclastogenesis by inducing cell cycle arrest, and *Cited2$^{-/-}$* cells displayed high levels of cell cycle genes and failed transition from proliferating pre-OCs to committed pre-OCs[22]. Here, we validated that loss of *Arid1a* could also lead to decreased expression of *Cited2*, which could further contribute to the above defective osteogenesis. Besides, our previous study on the functional mechanism of ARID1A in the fate commitment of mesenchymal stem cells and their progeny has shown that ARID1A maintains tissue homeostasis through limiting proliferation, promoting cell cycle exit and differentiation of transit-amplifying cells by inhibiting the *Aurka-Cdk1* axis[10]. Therefore, ARID1A could be intimately involved in the cell-cycle regulation during osteoclastogenesis as well, which needs to be explored in further studies.

The mammalian BAF complex is highly polymorphic, assembling its subunits with cell type- or developmental stage-specific character[38–41]. For example, an embryonic stem cell complex regulates pluripotency[51]; a neural precursor and neural BAF complexes fine-tune neurogenesis[40,52]; and BAF complex subunits shift for neural and muscle differentiation[40,52–55]. The variety of subunits assembled in distinct BAF complexes reflect the evolutionary diversification and biological complexity of the organisms. BRD9-ncBAF complex was recently discovered in mouse embryonic stem cells, which prevents developmental transition to a primed epiblast state by regulating pluripotency factors[42]. Based on a comparison of distinct genomic binding characters, Gatchalian et al. postulated the likelihood that cooperation may exist between ncBAF and esBAF complex to ensure successful naïve pluripotency maintenance[42]. Succeeding to our recent study, which demonstrated that BRD9-ncBAF complex negatively regulates OC differentiation through STAT1/IFN-β signaling[17], in our present study, the antagonist between ARID1A-cBAF and BRD9-ncBAF complex during cell fate canalization of osteoclastogenesis has been further validated with inhibitor assay and the compound mutant mouse model. Furthermore, our preliminary data showed the downregulated E3 ubiquitin-protein ligase MDM2 may contribute to the increased BRD9 protein level after loss of ARID1A. While it is still possible that an increased assembled form of BRD9 in the ncBAF complex when the ARID1A-cBAF complex is disrupted, may protect

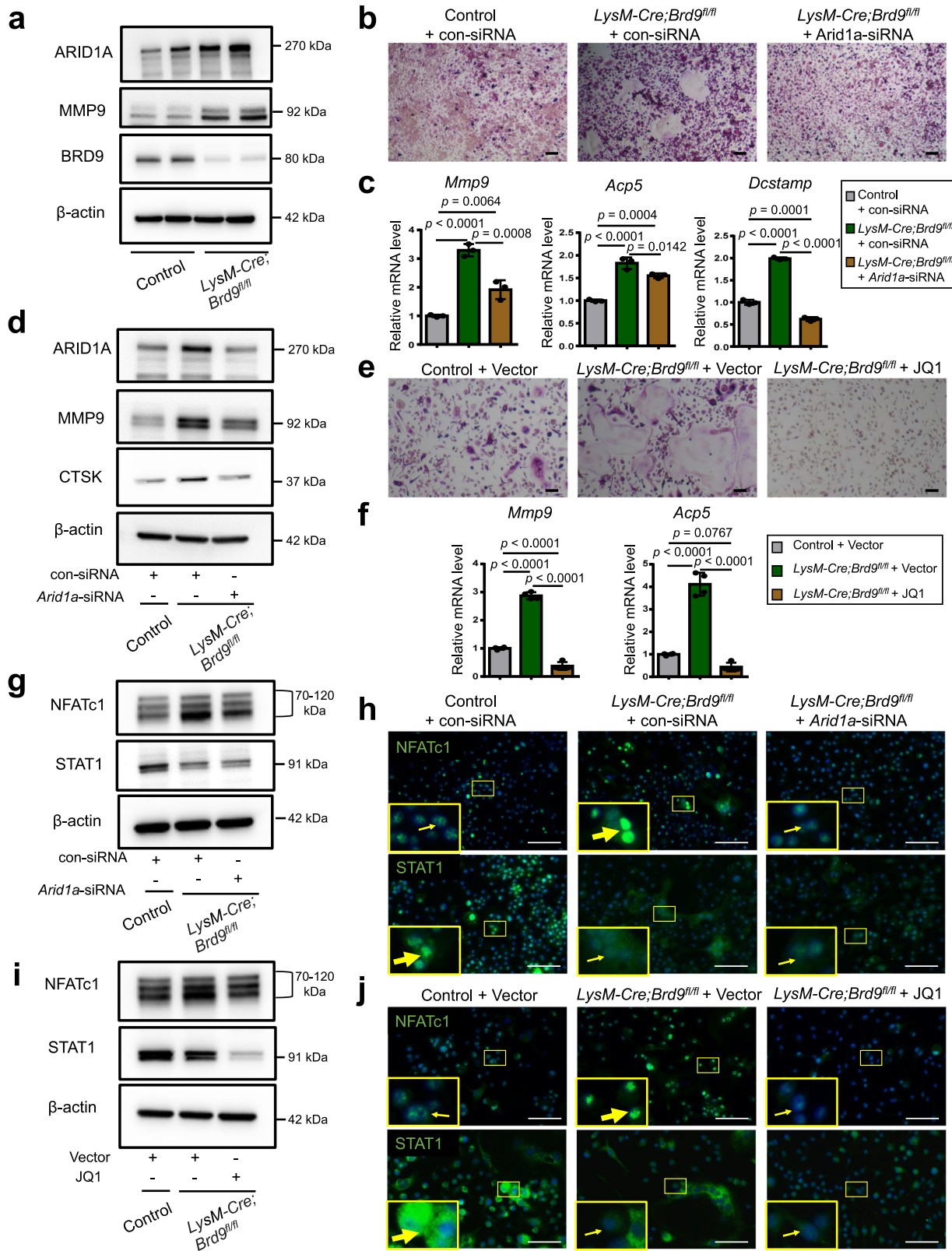

BRD9 from degradation per se, which needs further investigation. Studies into the significance of BRD9-ncBAF complex, and its cooperation or antagonist with ARID1A-cBAF complex in mammalian development are still limited. Whether it's universal or not, our findings provide meaningful clues for a further comprehensive understanding of diverse BAF complexes during developmental processes and disease pathologies.

It has been shown that the BRD9-ncBAF complex cooperates with BRD4 in naïve pluripotency maintenance in embryonic stem cells and macrophage activation[42,45]. Our previous study demonstrated the function specificity of BRD9 distinguished from BRD4 based on the distinct phenotypes and transcription profile change after BRD4 inhibition with that after BRD9 degradation/deletion in osteoclastogenesis[17]. In the present study, it is further validated that

**Fig. 6 | Enhanced activity of ARID1A/BRD4/PU.1/NFATc1 axis contributes to the overactivated OC lineage determination after loss of *Brd9*. a** The protein expression of ARID1A, MMP9, and BRD9 in BMCs from 4-week-old male *LysM-Cre;BRD9<sup>fl/fl</sup>* mice and littermate control mice after RANKL-induction, as measured by western blot. **b** TRAP staining of *Arid1a*-siRNA or control-siRNA treated BMCs from 4-week-old male *LysM-Cre;BRD9<sup>fl/fl</sup>* mice and littermate control mice after RANKL-induction. Scale bar, 100 μm. **c** The mRNA expression of *Mmp9, Acp5,* and *Dcstamp* in *Arid1a*-siRNA or control-siRNA treated BMCs from 4-week-old male *LysM-Cre;BRD9<sup>fl/fl</sup>* mice and littermate control mice after RANKL-induction, as measured by qPCR. *n* = 3 biologically independent samples. **d** The protein expression of ARID1A, MMP9, and CTSK in *Arid1a*-siRNA or control-siRNA treated BMCs from 4-week-old male *LysM-Cre;BRD9<sup>fl/fl</sup>* mice and littermate control mice after RANKL-induction, as measured by western blot. **e** TRAP staining of JQ1 or control vector treated BMCs from 4-week-old male *LysM-Cre;BRD9<sup>fl/fl</sup>* mice and littermate control mice after RANKL-induction. Scale bar, 100 μm. **f** The mRNA expression of *Mmp9* and *Acp5* in JQ1 or control vector treated BMCs from 4-week-old male *LysM-Cre;BRD9<sup>fl/fl</sup>* mice and littermate control mice after RANKL-induction, as measured by qPCR. *n* = 4 biologically independent samples. **g** The protein expression and **h** immunofluorescence of NFATc1 or STAT1 in *Arid1a*-siRNA or control-siRNA treated BMCs from 4-week-old male *LysM-Cre;BRD9<sup>fl/fl</sup>* mice and littermate control mice after RANKL-induction. Yellow arrows indicate positive signals. Scale bar, 100 μm. **i** The protein expression and **j** immunofluorescence of NFATc1 or STAT1 in JQ1 or control vector treated BMCs from 4-week-old male *LysM-Cre;BRD9<sup>fl/fl</sup>* mice and littermate control mice after RANKL-induction. Yellow arrows indicate positive signals. Scale bar, 100 μm. All data in this figure are represented as mean ± SD. One-way ANOVA with Tukey's multiple comparisons test for **c** and **f**. All experiments were performed in triplicates unless otherwise stated. Source data and exact *p* values are provided in the Source data file.

the antagonistic function of ARID1A-"accelerator" and BRD9-"brake" both depend on coactivator BRD4-"clutch" during cell fate canulization. These findings enlighten us that, differing from the definite role of chromatin remodeler ARID1A or BRD9, BRD4 could act as a cooperator participating in multiple critical transcription regulation activities. Undoubtedly, the pursuit of research in this field is continuing intriguing interest.

Abnormal osteoclastogenesis and activation are involved in multiple skeletal disorders, like osteoporosis, Paget's disease, rheumatoid arthritis, periodontitis, osteopetrosis, and so on[56]. Obtaining a better understanding of osteoclastogenesis is therefore central to the development of new treatments for bone diseases. This study provides a comprehensive picture of the function of cBAF and ncBAF complex during OC differentiation trajectory, opening avenues for future investigation into the physiology and pathology of the skeletal and immune systems. SEs, ever since identified, have been considered promising therapeutic targets in cancer and other diseases[57,58]. While, there are limited studies yet to indicate whether SE inhibitors can be used for the potential treatment of bone-related diseases, mainly due to the insufficient mechanism research on how SEs are regulated and the detailed molecular mechanism of how SEs regulate their target genes. This study establishes that the ARID1A/BRD4/*Nfatc1* SE axis in OC represents a potential therapeutic target for bone disease. While how to achieve SE-targeted therapy for bone-related diseases in clinical remains to be investigated deeply. Overall, this study provides insights into the therapeutic potential of chromatin remodeling for bone disease and expands our knowledge of sophisticated cooperation between chromatin remodelers, coactivators, and lineage-specifying TF at master TF SEs in a condensate manner, and also antagonists between distinct BAF complexes in the proper and balanced cell fate canalization.

## Methods

### Mice
*LysM-Cre* mice (Strain NO. T003822), *Arid1a-flox* mice (Strain NO. T013487) *Brd9-flox* mice (Strain NO. T008489), and WT C57BL/6 J mice (Strain NO. N000013) from GemPharmatech (Nanjing, China) and tdTomato (Strain NO. 007909) mouse line[59] from Jackson Laboratory (Maine, USA) were used and cross-bred in this study. All mice were used for analysis regardless of sex. All mice were housed in pathogen-free conditions with constant ambient temperature ($22 \pm 2\,°C$) and humidity ($55 \pm 10\%$), with an alternating 12-hour light/dark cycle. All mice were euthanized by carbon dioxide overdose followed by cervical dislocation. All animal studies were approved by the Institutional Animal Care and Use Committee at Ninth People's Hospital, School of Medicine, Shanghai Jiao Tong University (SH9H-2022-A926-1).

### Micro-CT analysis
Micro-CT (μCT) analysis of fixed femur was performed using a Skyscan 1176 (Bruker, Kontich, Belgium) at 50 kVp, 450 uA, and a resolution of 9 μm. The obtained images were reconstructed with NRecon software

(v1.7.1.0, Bruker, Kontich, Belgium). The region from 50 to 250 slices below the growth plate was analyzed for BV/TV, Tb.Th., Tb.N., Tb.Sp, BMD, and Po and the region from 500 to 550 slices below the growth plate was analyzed for B.Th and B.Pm using the program CTAn (v 1.16, Bruker, Kontich, Belgium).

### ELISA
The concentrations of serum TRAP, CTX-I, OCN, and P1NP proteins were measured by ELISA (Elabscience Biotechnology Co., Ltd. E-EL-M1116, E-EL-M3023, E-EL-M0864, and E-EL-M0233). All the procedures were performed following the manufacturer's instructions.

### Histological analysis
For decalcification, mouse femurs were dissected and fixed, followed by decalcification in 10% EDTA in PBS for 1-3 weeks depending on the age of the samples. The decalcified samples were dehydrated with serial ethanol and xylene and embedded in paraffin. The paraffin-embedded samples were then cut into sections with a thickness of 4 mm using a microtome (Leica). H&E staining (Servicebio, G1005), TRAP Staining (Sigma, 387 A), toluidine blue staining (Servicebio, G1032), and safranin O/fast green staining (solarbio, G1371) were conducted following the manufacturer's standard protocol. Referring to the RGB trichrome staining method described in a publication[21], sections were stained sequentially with 1% alcian blue 8 G (Sangon Biotech, A600298) for 20 min and 0.04% fast green FCF (Sangon Biotech, A610452) for 20 min. Finally, the sections were stained with 0.1% sirius red (MKBio, MM1036) for 30 min. As for von Kossa staining (servicebio, G1043), uncalcified bone tissues were dehydrated and embedded in an OCT compound (Tissue-Tek, Sakura) before being sectioned at 8 mm using a cryostat (Leica CM1850) and adhesive films (cryofilm type 3 C(16UF) 2.5 cm, Section lab, Hiroshima, Japan). Zeiss Axio Scope A1 Microscope (ZEISS, German) with DP2-TWAIN software (v3.0.0.6212) and Leica TCS SP8 Confocal Microscope (Leica microsystems, Germany) were used to acquire images.

### Immunofluorescence staining
The decalcified samples were dehydrated in serial sucrose solutions and embedded in an OCT compound. OCT-embedded samples were cryosectioned at 8 mm using a cryostat followed by staining. For immunofluorescence staining, cryosections were soaked in a blocking solution for one hour at room temperature and then incubated with primary antibodies diluted in a blocking solution at 4 °C overnight. After washing three times in PBS, the sections were incubated with alexa-conjugated secondary antibodies and counterstained with DAPI. Antibodies used in immunofluorescence staining are as following: ARID1A antibody (Cell signaling, 12354, 1:100; novusbio, NBP2-61623, 1:100), CTSK antibody (Abcam, ab37259, 1:100), ACP5 antibody (Abcam, ab235448, 1:100), NFATc1 antibody (Bioworld, bs6677, 1:100), BRD4 antibody (Bethyl Laboratories, A700-004, 1:100), PU.1 antibody (abnova, H00006688-M02, 1:100), STAT1 antibody (Cell signaling,

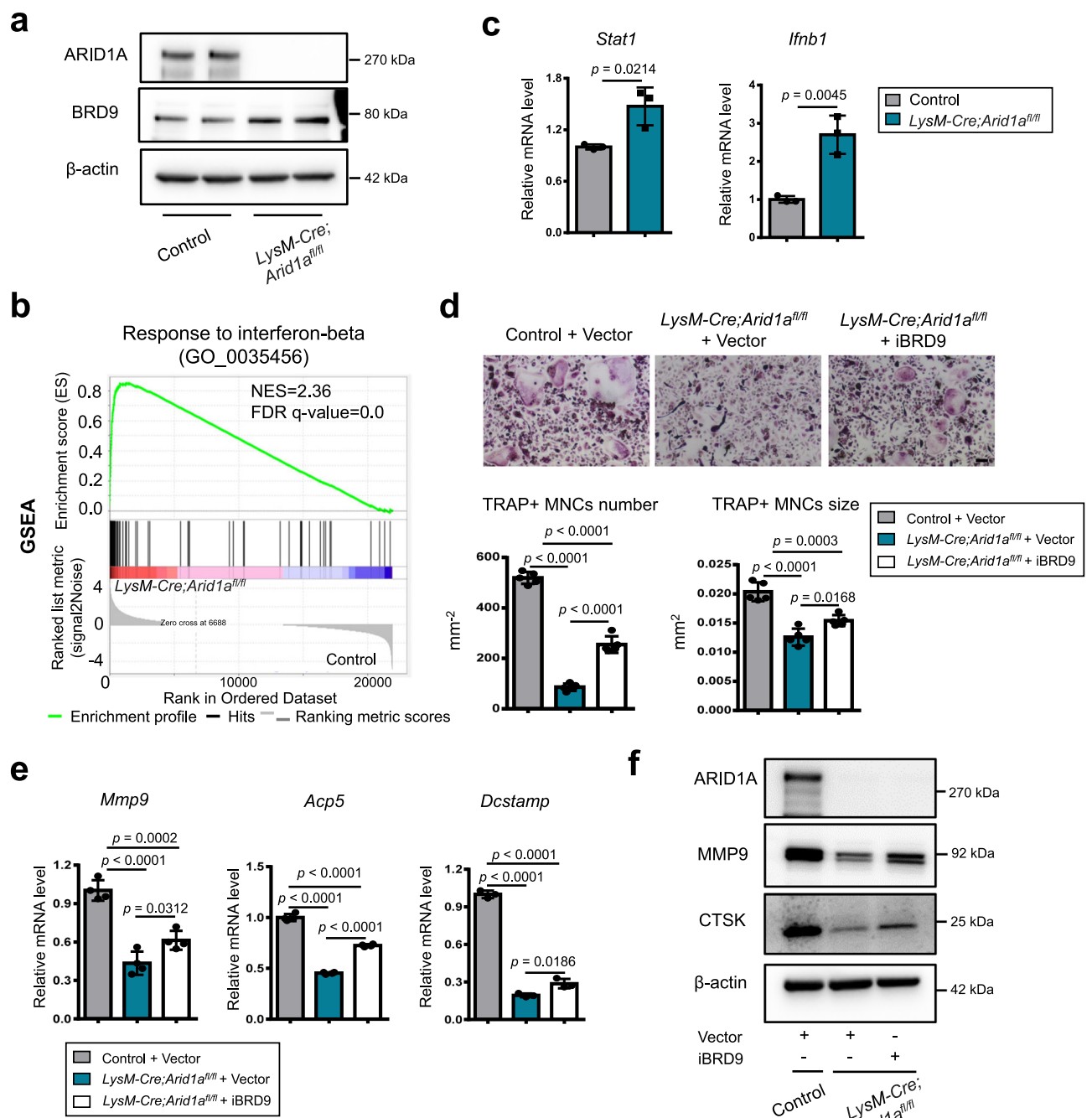

**Fig. 7 | Excessive activity of BRD9/STAT1 axis contributes to the compromised OC differentiation process after loss of *Arid1a*. a** The protein expression of ARID1A and BRD9 in BMCs from 4-week-old male *LysM-Cre;Arid1a^fl/fl^* mice and littermate control mice after RANKL-induction, as measured by western blot. **b** GSEA analysis of DEG profiles in bulk RNA-seq between the control group and *LysM-Cre;Arid1a^fl/f^* group. **c** The mRNA expression of *Stat1* and *Ifnb1* in BMCs from 4-week-old male *LysM-Cre;Arid1a^fl/fl^* mice and littermate control mice after RANKL-induction, as measured by qPCR. *n* = 3 biologically independent samples. **d** TRAP staining and quantification analysis of iBRD9 or control vector treated BMCs from 4-week-old male *LysM-Cre;Arid1a^fl/fl^* mice and littermate control mice after RANKL-induction. Scale bar, 100 μm. *n* = 5 biologically independent samples. **e** The mRNA

expression of *Mmp9, Acp5 and Dcstamp* and in iBRD9 or control vector treated BMCs from 4-week-old male *LysM-Cre;Arid1a^fl/fl^* mice and littermate control mice after RANKL-induction. *n* = 4 biologically independent samples for *Mmp9* and *Acp5*. *n* = 3 biologically independent samples in *Dcstamp*. **f** The protein expression of ARID1A, MMP9, and CTSK in iBRD9 or control vector treated BMCs from 4-week-old male *LysM-Cre;Arid1a^fl/fl^* mice and littermate control mice after RANKL-induction, as measured by western blot. All data in this figure are represented as mean ± SD. Two-tailed Student's *t*-test for **c**. One-way ANOVA with Tukey's multiple comparisons test for **d** and **e**. All experiments were performed in triplicates unless otherwise stated. Source data and exact *p* values are provided in the Source data file.

9172, 1:100), goat anti-rabbit IgG (H + L) cross-adsorbed secondary antibody, alexa fluor 488 (Thermo Fisher Scientific, A-11008, 1:200), donkey anti-rabbit IgG (H + L) highly cross-adsorbed secondary antibody, alexa fluor 594 (Thermo Fisher Scientific, A-21207, 1:200), goat anti-mouse IgG (H + L) cross-adsorbed secondary antibody, alexa fluor

488 (Thermo Fisher Scientific, A-11001, 1:200), goat anti-mouse IgG (H + L) cross-adsorbed secondary antibody, alexa fluor 594 (Thermo Fisher Scientific, A-11005, 1:200), goat anti-mouse IgG (H + L) cross-adsorbed secondary antibody, alexa fluor™ 647 (Thermo Fisher Scientific, A-21235, 1:200). Nikon Eclipse Ti-U Microscope (Nikon, Japan)

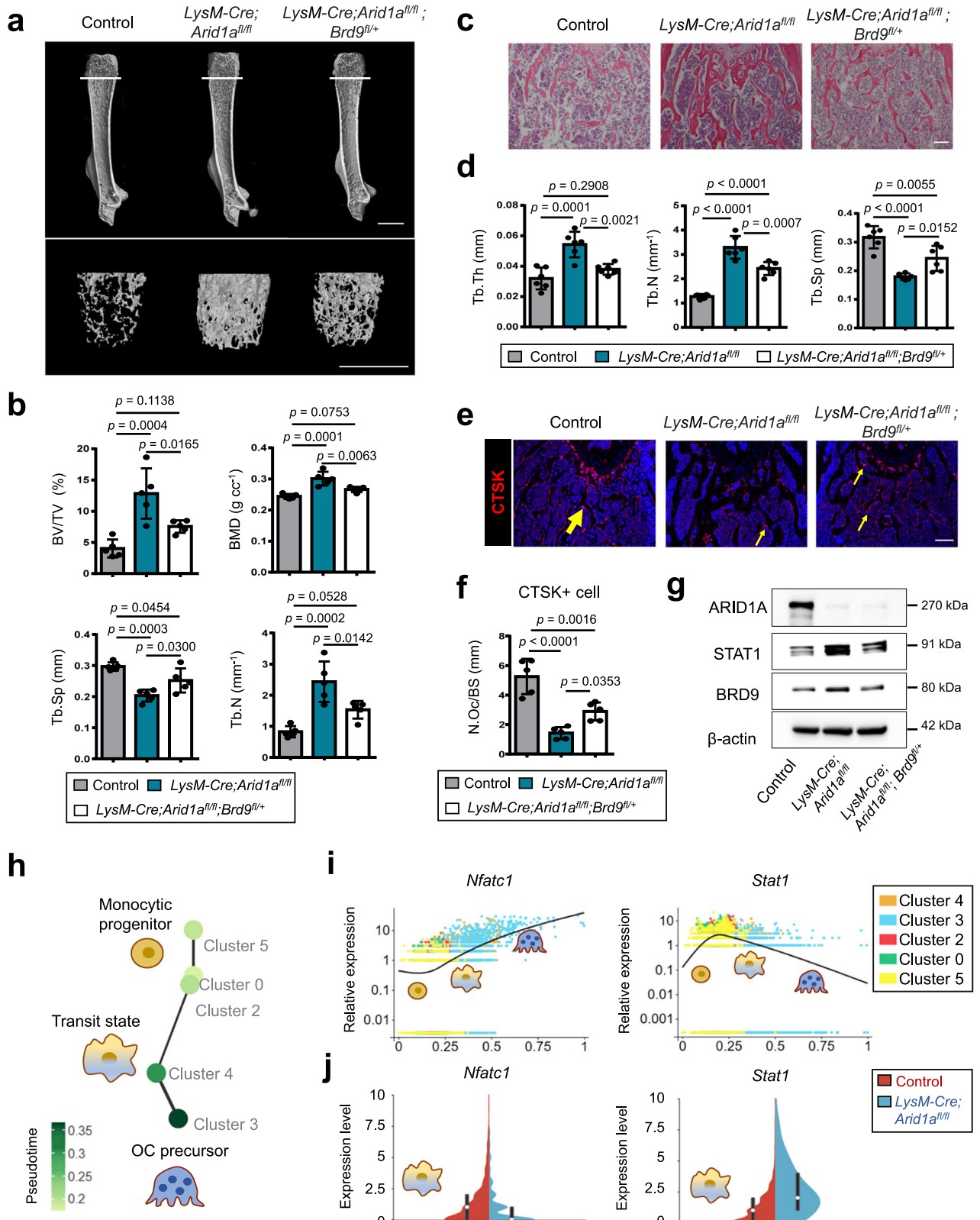

with NIS-Elements software (v4.5000.1117.0), Zeiss Axio Scope A1 Microscope with DP2-TWAIN software, and Leica TCS SP8 Confocal Microscope were used to acquire images.

## Cell culture

BMCs were harvested from mouse femurs of four to six-week-old mice and cultured in α-minimum essential medium (α-MEM) (Thermo Fisher Scientific, Waltham, MA, USA) containing 10% FBS (OriCell® FBSAD-01011-500, Cyagen Biosciences, Guangzhou, China) and 1% penicillin-streptomycin overnight (Thermo Fisher Scientific, Waltham, MA, USA). For osteoclastic induction, BMCs were cultured into complete media with 50 ng/ml recombinant soluble murine M-CSF (PeproTech, 315-02) and 100 ng/ml recombinant soluble murine RANKL (PeproTech, 315-11). For the inhibitor experiments, the cells were treated with either

**Fig. 8 | *Brd9* heterozygous knockout partially rescues OC differentiation defects after loss of *Arid1a* in vivo. a** Representative micro-CT image and **b** quantification analysis of bone volume/tissue volume ratio (BV/TV), bone mineral density (BMD), trabecular separation (Tb.Sp), and trabecular number (Tb.N) of femurs from 3-month-old female *LysM-Cre;Arid1a$^{fl/fl}$* mice, *LysM-Cre;Arid1a$^{fl/fl}$;BRD9$^{fl/}$*$^{+}$ mice, and littermate control mice. Distal trabecular bone are enlarged on the lower panel. Scale bar, 2 mm. *n* = 5. **c** H&E staining and **d** quantification analysis of trabecular thickness (Tb.Th), Tb.N, and Tb.Sp of the distal femoral trabecular bone from 3-month-old female *LysM-Cre;Arid1a$^{fl/fl}$* mice, *LysM-Cre;Arid1a$^{fl/fl}$;BRD9$^{fl/+}$* mice, and littermate control mice. Scale bar, 200 μm. *n* = 6. **e** CTSK immunofluorescence (red) staining and **f** quantification analysis of the distal femoral trabecular bone from 3-month-old female *LysM-Cre;Arid1a$^{fl/fl}$* mice, *LysM-Cre;Arid1a$^{fl/fl}$;BRD9$^{fl/+}$* mice,

and littermate control mice. *n* = 5. Scale bar, 200 μm. Yellow arrows indicate a positive signal. **g** The protein expression of ARID1A, STAT1, and BRD9 in BMCs from 4-week-old male *LysM-Cre;Arid1a$^{fl/fl}$* mice, *LysM-Cre;Arid1a$^{fl/fl}$;BRD9$^{fl/+}$* mice, and littermate control mice after RANKL-induction, as measured by western blot. **h** PAGA graph of cell clusters showing trajectory colored by the timeline and **i** the expression patterns of *Nfatc1* and *Stat1* during the trajectory of osteoclastogenesis. **j** bean plot of the different gene densities of *Nfatc1* and *Stat1* in cluster 4 between the control group and *LysM-Cre;Arid1a$^{fl/fl}$* group. All data in this figure are represented as mean ± SD. One-way ANOVA with Tukey's multiple comparisons test for **b**, **d**, and **f**. All experiments were performed in triplicates unless otherwise stated. Source data and exact *p* values are provided in the Source data file.

vehicle or iBRD9 (1 μM, Tocris Bioscience, 5591), JQ1 (0.08 μM, Abmole Bioscience, M2167), and SP-141 (0.06 μM, Csnpharm, 223123) twenty-four hours after plated. For cell cultures, media was changed every two days until indicated time points.

### EdU incorporation and detection
For EdU incorporation and detection, mutant mice and littermate controls were injected with EdU (10 mg/g body weight, Thermo Fisher, A10044) intraperitoneally 3 hours before being euthanized. The femurs were fixed and decalcified. For cultured cell, the cells were treated with EdU (10 μM) for twelve hours before being fixed. Click-iT plus EdU cell proliferation kit (Thermo Fisher, C10637) was used for in situ EdU detection according to the manufacturer's instructions. Nikon Eclipse Ti-U Microscope with NIS-Elements software and Leica TCS SP8 Confocal Microscope were used to acquire images.

### Lentivirus and siRNA transfection
*Nfatc1*-overexpression lentivirus (*HBLV-m-Nfatc1-ZsGreen-PURO*) with its control lentivirus (*HBLV-ZsGreen-PURO*) were purchased from Hanbio Biotechnology Co. Ltd (Shanghai, China). For lentivirus transfection, 2 × 10$^5$ BMCs were seeded in 12-well plates in α-MEM overnight and were infected with lentivirus expressing either target gene or control lentivirus at a multiplicity of 1. 48 hours after transfection, the BMCs were induced with RANKL for osteoclastic induction. After 24 hours of osteoclastic induction, the cultured cells were harvested for evaluation. For siRNA transfection, Arid1a siRNA (QIAGEN, 1027418, Mm_Arid1a_5 FlexiTube siRNA SI02676058) and control siRNA (Thermo Fisher, 4390844), lipofectamineTM RNAiMAX (Thermo Fisher, 13778075) and Opti-MEM I Reduced Serum Medium (Thermo Fisher, 31985062) were used in this study. The final concentration of each siRNA was 25 nM. 48 hours after transfection, the BMCs were induced with RANKL for osteoclastic induction. After 24 hours of osteoclastic induction, the cultured cells were harvested for evaluation.

### Quantitative reverse transcription PCR (qPCR)
For qPCR analysis, RNA was extracted using TRIzol (Sigma, T9424) and was reverse-transcribed with the Prime Script RT master kit (TaKaRa Bio Inc., RR036A). Then the relative amounts of each mRNA transcript were analyzed using the Roche LightCycler 480 system (v1.5.1.74) with SsoAdvanced Universal SYBR Green Supermix (Bio-Rad, 1725270). The expression of β-actin as an internal control. Primer sequences are listed in Supplementary Data 4.

### Single-cell RNA-sequencing analysis
Single-cell RNA-seq was performed on cells at 24 h post-RANKL induction. Single cells were loaded on a 10X Genomics GemCode Single-cell instrument. Libraries were generated and sequenced from the cDNAs with Chromium Next GEM Single Cell 3' Reagent Kits v3.1. and subjected to Illumina NovaSeq 6000. Quality control, alignment, quantification, and aggregation of sample count matrices were performed using the 10x Genomics Cell Ranger pipeline according to the

manufacturer's protocol. Significantly upregulated genes were identified with at least 1.28-fold overexpressed in the target cluster, expressed in more than 25% of the cells belonging to the target cluster and the *p*-value is less than 0.01. The differentially expressed genes in the given cluster were further conducted with KEGG pathway enrichment analysis[60].

Pseudotime analysis was conducted using monocle2, a method that can predict the differentiation trajectory based on an individual cell's asynchronous progression of the process with a standard protocol with default parameters. The estimation of the root of the trajectory was based on the cluster identities and marker gene analysis[22]. Partition-based graph abstraction (PAGA), which allows robust reconstruction branching gene expression changes across different datasets, was also conducted according to standard protocol[61]. Correlation statistics and significance calculations for each gene were conducted using the Rfast2 software package. Then the significantly differential expression genes with FDR <1e-7 and genes with similar trends in expression were identified on the pseudotime axis.

### RNA-sequencing analysis
For RNA-sequencing analysis, libraries were prepared using the NEB-Next Ultra II RNA Library Prep Kit and then sequenced on the Illumina NovaSeq 6000 platform. Raw reads were filtered using Cutadapt (v1.15) and aligned with the GRCm39 genome using HISAT2 v2.0.5. Read Count values on each gene were compared using HTSeq (v0.9.1) and normalized to FPKM. Then difference expression of genes was analyzed using DESeq (v1.30.0) (fold change >= 2 and *p* < 0.05). GSEA analysis was performed to functionally annotate the relevant genes and assess the enriched signaling pathways using GSEA_Linux_4.1.0.

### Western blot and co-immunoprecipitation (co-IP)
For western blot analysis, total protein was obtained and homogenized in RIPA buffer (Cell Signaling, 9806 s) supplemented with protease inhibitor (Thermo Fisher Scientific, A32959). Western blot was performed per standard protocol and signals were detected using the UVITEC Alliance system (v16.0.3.0). Antibodies used in western blot as following: ARID1A antibody (Cell signaling, 12354, 1:1000), MMP9 antibody (Abcam, ab228402, 1:1000), CTSK antibody (Abcam, ab37259, 1:1000), NFATc1 antibody (BioLegend, 649601, 1:500), BRD4 antibody (Bethyl Laboratories, A700-004, 1:1000), PU.1 antibody (Abcam, ab227835, 1:1000), BRD9 antibody (Abcam, ab259839, 1:1000), STAT1 antibody (Cell signaling, 9172, 1:1000), β-actin antibody (Abcam, ab20272HRP, 1:5000), mouse IgG HRP-conjugated antibody (R&D, HAF007, 1:1000) and rabbit IgG HRP-conjugated antibody (R&D, HAF008, 1:1000).

For co-IP, BMCs after osteoclastic induction were harvested and lysed in Pierce™ IP Lysis Buffer (Thermo Fisher Scientific, 87787). Then lysates were subjected to immunoprecipitation with ARID1A antibody (Cell signaling, 12354, 1:100), BRD9 antibody (Bethyl Laboratories, A700-153, 1:100), or normal Rabbit IgG (Cell Signaling, 2729, the same concentration as the specific target antibody) and Dynabeads™ Protein A (Thermo Fisher Scientific, 10001D). Immune complexes were

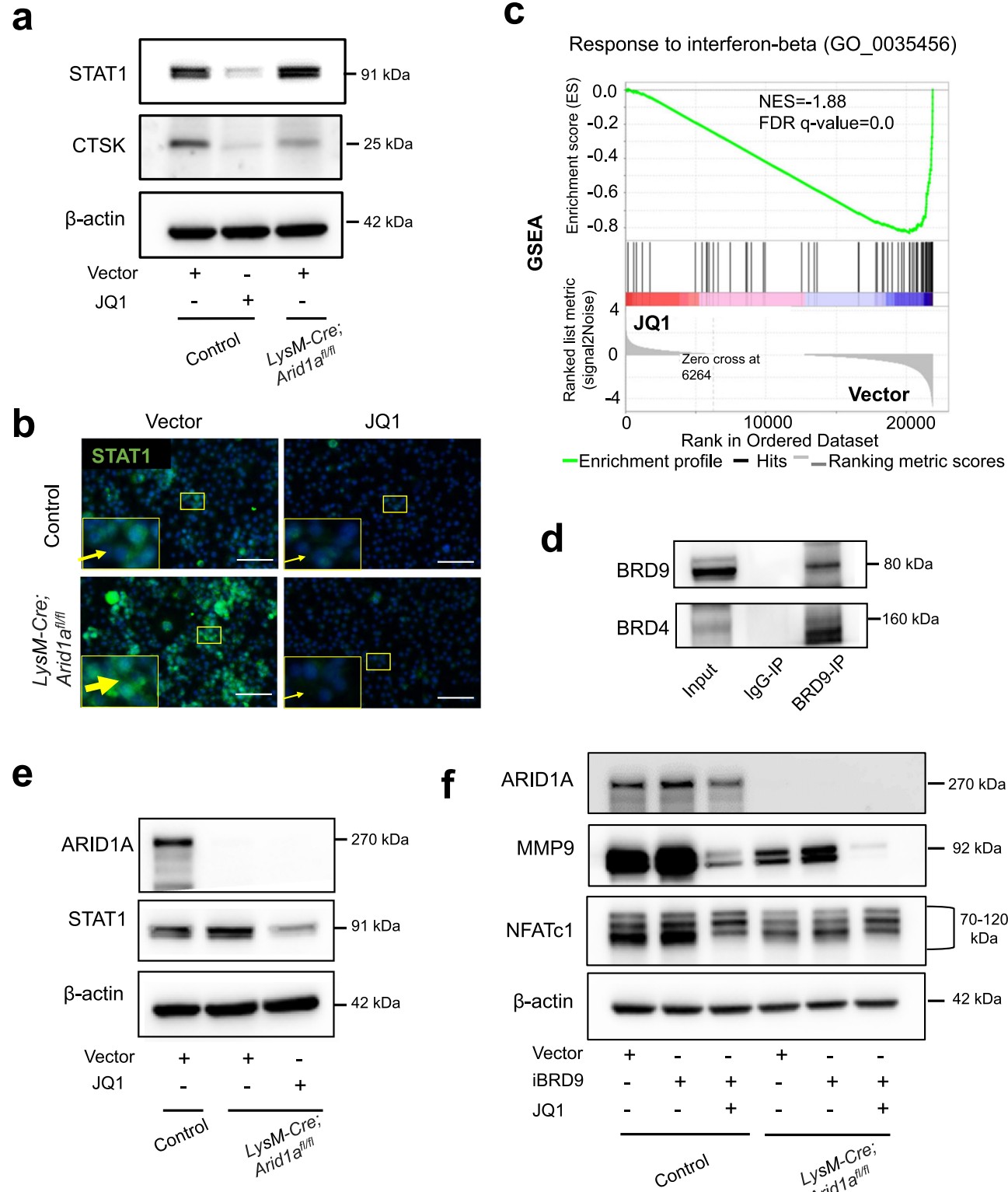

**Fig. 9 | The antagonistic function of ARID1A and BRD9 both depend on BRD4 during osteoclastogenesis. a** The protein expression of STAT1 and CTSK and **b** STAT1 immunofluorescence (green) in JQ1 or control vector treated BMCs from 4-week-old male *LysM-Cre;Arid1a^fl/fl* mice and littermate control mice after RANKL-induction. **c** GSEA analysis of DEG profiles in bulk RNA-seq between JQ1 or control vector treated BMCs after RANKL-induction. **d** Co-IP assay with BRD9 antibody (or IgG) in BMCs during osteoclastic induction, followed by immunoblotting of BRD4 and BRD9. **e** The protein expression of ARID1A and STAT1 in JQ1 or control vector treated BMCs from 4-week-old male *LysM-Cre;Arid1a^fl/fl* mice and littermate control mice after RANKL-induction. **f** The protein expression of ARID1A, MMP9 and NFATc1 in JQ1, iBRD9 or control vector treated BMCs from 4-week-old male *LysM-Cre;Arid1a^fl/fl* mice and littermate control mice after RANKL-induction. All experiments were performed in triplicates unless otherwise stated. Source data are provided in the Source data file.

washed and subjected to immunoblotting with ARID1A antibody (Cell signaling, 12354, 1:1000), BRD4 antibody (Bethyl Laboratories, A700-004, 1:1000), PU.1 antibody (Abcam, ab227835, 1:1000), BRD9 antibody (Abcam, ab259839, 1:1000).

## ChIP assay and analysis
BMCs before and after 24 h RANKL induction were fixed with formaldehyde for further ChIP-sequencing analysis or qPCR. The Simple ChIP Plus Enzymatic Chromatin IP Kit (Magnetic Beads; Cell Signaling Technology 9005), H3K27Ac antibody (Cell signaling, 8173, 1:100), ARID1A antibody (Cell signaling, 12354, 1:100), BRD4 antibody (Bethyl Laboratories, A700-004, 1:100), PU.1 antibody (Abcam, ab227835, 1:100) with normal rabbit IgG (Cell Signaling, 2729, the same concentration as the specific target antibody) as a non-specific IgG control were used following the manufacturer's instructions.

ChIP-sequencing libraries were prepared using the KAPA HTP Library Preparation Kit and were sequenced on the Illumina Novaseq 6000 system. Double-end sequencing was used with a 150 bp read length. Raw reads were filtered using FASTX-Toolkit (v0.0.14) (http://hannonlab.cshl.edu/fastx_toolkit/). Then the clean reads were aligned to the GRCm39 mouse genome using Bowtie 2 (v2.3.5.1)[62]. Multiple aligned reads were filtered out using Picard (v2.1.0) (http://broadinstitute.github.io/picard/). Then the unique mapped reads without duplicated reads were called for peaks using MACS2 (v2.1.0)[63] with default parameters and $p < 0.05$. The peaks were annotated using ChIPSeeker (v1.20.0)[64]. For motif enrichment analysis we used AME with the same background model and negative control[65]. AME tool of the MEME suite provides motifs that are enriched compared to the shuffled background[65]. We set the E-value threshold (E-value < =10, Fisher's exact test) for reporting enriched motifs. We used motif database JASPAR2022_CORE_vertebrates_non-redundant_v2. meme to test for enrichment.

Active enhancers were defined as regions of ChIP-seq enrichment for H3K27Ac outside of promoters (e.g., a region not contained within ±2.5 kb region flanking the promoter). To accurately capture dense clusters of enhancers, we allowed H3K27Ac regions within 12.5 kb of one another to be stitched together. To identify SEs, we first ranked all enhancers by increasing the total background subtracted ChIP-seq-occupancy of H3K27Ac (x-axis) and plotted the total background subtracted ChIP-seq occupancy of H3K27Ac in units of total rpm (y-axis). This representation revealed a clear inflection point in the distribution of H3K27Ac at enhancers. We geometrically defined the inflection point and used it to establish the cutoff for SEs. Using a simple proximity rule, we assigned all transcriptionally active genes (TSSs) to SEs within a 50 kb window[24]. Sequences of the *Nfatc1* ChIP primers at SE are listed in Supplementary Data 4.

## Dual luciferase reporter assays
The Nfatc1 promoter region[29] (*Pro*, 0.8 kb, Chr18:80,756,240-80,757,039), enhancer region 1 (*E1*, 0.59 kb, Chr18:80,758,246-80,758,838), enhancer region 2 (*E2*, 0.71 kb, Chr18:80,745,769-80,746,481) and enhancer region 3 (*E3*, 0.34 kb, Chr18:80,734,924-80,735,265) was synthesized and cloned into *pGL3*-basic luciferase reporter as and *pGL3-Nfatc1-Pro, pGL3-Nfatc1-Pro-E1, pGL3-Nfatc1-E2-Pro, pGL3-Nfatc1-E3-Pro* and *pGL3-Nfatc1-E3-E2-Pro-E1*. For transfection, $2 \times 10^5$ Raw264.7 cells were seeded in 24-well plates in α-MEM overnight and were infected with above reporter clones (400 ng) and Renilla luciferase plasmids (pRL-TK) (200 ng) using Lipofectamine 3000 transfection reagent (Thermo Fisher Scientific, L3000008) and Opti-MEM™ I Reduced Serum Medium (Thermo Fisher Scientific, 31985062) according to the manufacturer's protocol. After 24 h incubation, the transfected Raw264.7 cells were osteoclastic induced with 100 ng/ml RANKL. After another 24 h incubation, the cell was lysed for a dual luciferase reporter gene assay kit (Beyotime, RG027) according

to the manufacturer's instructions. Renilla luciferase is used as an internal standard control, and the normalized reporter activity of all samples was compared. Each group had six replicate samples. Independent experiments were repeated in triplicate.

## 3C-quantitative polymerase chain reaction
3C-qPCR was performed as previously described with some modification[12,66,67]. 3 C libraries for BMCs before and after 24 h RANKL induction were prepared following the cross-linking with fresh 1% formaldehyde at room temperature for 10 minutes with mixing. To quench the crosslinking reaction, the samples were incubated with glycine at a final concentration of 0.125 M on ice for at least 15 min. After centrifugation, the pellet was incubated in ice-cold lysis buffer (10 mM Tris-HCl, pH 8.0; 10 mM NaCl; 0.2% IGEPAL® CA-630) containing 1x protease inhibitor (Roche 1836145001) on ice for 60 min. After centrifugation, the pellet was incubated in restriction enzyme buffer containing 0.3% SDS for 1 h at 37 °C while shaking at 900 rpm. Then add triton X-100 (final: 2%) and incubate for 1 h at 37 °C while shaking at 900 rpm. Then, the chromatin was digested with 200U of restriction enzyme DpnII (NEB R0543S) overnight at 37 °C while shaking. After inactivating the restriction enzyme, it was followed by random re-ligation with 100 U T4 DNA Ligase (Thermo Fisher Scientific, 15224025) and incubated overnight at 16 °C. After reversing crosslinks by incubation with proteinase K (0.1 mg ml $^{-1}$) at 65 °C for 4 h, DNA was purified by phenol/chloroform extraction and ethanol precipitation. 3 C libraries were quantified with Qubit dsDNA Quantitation Assay kits (Thermo Fisher Scientific, Q32854) and diluted to the same concentration to ensure an equal amount of input for qPCR, which was performed using the Roche LightCycler 480 system (v1.5.1.74) with Hieff Unicon® qPCR TaqMan Probe Master Mix (Yeasen, 11205ES03). The short-range ligation products using the anchor primer in combination with a primer for adjacent DpnII fragments were used for normalizing the relative interaction frequency with the *Nfatc1* promoter among different samples to control for differences in cross-linking and ligation efficiencies as described previously[67]. Primer and probe sequences are listed in Supplementary Data 5.

## Statistical and reproducibility
All statistical analyses were performed with GraphPad Prism v6.01 software and are presented as mean ± standard deviation. The unpaired two-tailed t-test between two groups or one-way analysis of variance (ANOVA) with Tukey's or Dunnett's multiple comparisons post-hoc test among three or more groups was used for comparisons. $P < 0.05$ is considered statistically significant. All experiments were repeated in triplicate or more unless otherwise stated.

## Reporting summary
Further information on research design is available in the Nature Portfolio Reporting Summary linked to this article.

## Data availability
The scRNA, bulk mRNA, and ChIP-sequencing data generated in this study have been deposited in the Gene Expression Omnibus database under accession code GSE245258. GRCm39 genome is referenced in this study [http://asia.ensembl.org/Mus_musculus/Info/Index]. The other relevant data generated in this study are provided in the Supplementary Information/Source Data file. Source data are provided with this paper.

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

## Acknowledgements

This work was sponsored by funding from the National Natural Science Foundation of China (82201004 to J.D., 81921002 to X.J., 82130027 to X.J., 82100966 to G.Y., 31900971 to M.Z.), the Young Elite Scientists Sponsorship Program by CAST (YESS20230102 to J.D. and 2021QNRC001 to G.Y.), the innovative research team of high-level local universities in Shanghai (SHSMU-ZLCX20212400 to X.J.). We thank for the sequencing and analysis service provided by Shanghai LUOXI Healthcare Technology Co., Ltd., Shanghai Personal Biotechnology Co., Ltd., China, and Guangzhou Genedenovo Biotechnology Co., Ltd.

## Author contributions

J.D. and X.J. conceived and designed the study; J.D., Y.L., J.S., E.Y., J.X., X.W., L.X., M.Z., and G.Y. performed the experiments; J.D. provided critical resources; J.D., Y.L., and G.Y. analyzed the sequencing data; J.D., G.Y., and X.J. provided technical assistance and supervision; All authors contributed to the interpretation of experiments. J.D., Y.L., G.Y., and X.J. wrote the manuscript with input from all co-authors. All authors discussed the results and contributed to the manuscript.

## Competing interests

The authors declare no competing interests.
