## [Peer Review File · Nature Communications]

REVIEWER COMMENTS

Reviewer #1 (Remarks to the Author):

In this study, the authors showed that chromatin remodeler ARID1A critically contributes to the osteoclast fate determination. Using genetic loss-of-function experiments, genome-wide epigenetic analyses and single-cell RNAseq, the authors demonstrated that ARID1A activates NFATc1 expression by binding to the super enhancer region with BRD4 and PU.1. The experiments are well controlled, results are clear and the conclusion appears sound, however, there are several important points to be addressed.

1) The molecular mechanisms underlying NFATc1 induction have been well studied in osteoclast research field. In this study, the authors showed that the ARID1A/ BRD4/PU.1 condensate is a critical determinant of NFATc1 induction but previous studies have shown that NF- κ B, AP-1 and NFATc1 itself contribute to the NFATc1 induction by activating cis-regulatory elements. Do these factors physically associate with the ARID1A/ BRD4/PU.1 condensate and regulate the same super enhancer region? The authors should clarify the relationship between the ARID1A/ BRD4/PU.1 complex and other previously reported factors (NF- κ B, AP-1 and NFATc1) in the regulation of NFATc1 induction.

2) This reviewer feels that the ARID1A story is sufficiently novel and interesting but the BRD9/STAT1 story makes this paper very complicated. The concluding schema in Figure.8 is confusing and does not reflect the whole story or main message of this manuscript. The authors should remove or revise the schema and please clearly discuss how ARID1 complex and BRD9 complex regulate each other.

3) It is interesting that the authors' scRNAseq data suggested that ARID1a regulates the transition of proliferation-differentiation during osteoclastogenesis. Previous osteoclast scRNAseq papers showed Cited2 (Nat Metab. 12:1382-1390, 2020) and Rab38 (JBMR Plus . 16;6:e10631. 2022) also control the transition of proliferation-differentiation in osteoclasts. The authors should examine the expression levels of Cited2 and Rab38 in Arid1a KO osteoclasts in Fig.2g and discuss the possibility that ARID1a is involved in the cell-cycle regulation during osteoclastogenesis.

4) Why IFN- β signaling is activated in ARID1a KO osteoclasts (Fig7b,c)? Does this mean IFN- β production is elevated and this acts on osteoclasts in an autocrine manner? The authors should show the expression levels of IFN- β in ARID1a KO osteoclasts.

5) Quantification of bone histomorphometry is needed in Figure 2a-e and Figure 7 j,k.

6) The physical association of ARID1A/ BRD4/PU.1 should be confirmed by immunoprecipitation.

7) There are several sentences where adverbs should be used instead of adjectives (e.g. page 5 line 90; BRD9/ncBAF negative regulates → negatively, page 9 line 174; ARID1A transcriptional promotes → transcriptionally, page 12 line 227; are high enriched → highly)

Reviewer #2 (Remarks to the Author):

In this manuscript, Du et al show that ARID1A complex accelerates osteoclastogenesis, while BRD9 complex negatively regulates osteoclast differentiation. The authors focused on BRD4 as a molecule to determine cell fate canalization during osteoclastogenesis, which is colocalized with ARID1A and PU.1 on *Nfatc1* gene. The inhibition of BRD4 suppresses BRD9-KD induced osteoclast differentiation. The proposed pathway is interesting and unifies the previous finding (Du et al., 2023) about BRD9-mediated negative regulation of osteoclastogenesis. However, there are several major concerns about the putative mechanism. The biggest issue is that the proximal mechanism by which RANK signaling induces the enhancer-promoter proximity of *Nfatc1* gene that is dependent on ARID1A/BRD4/PU.1 complex formation is not clear.

Major comments

Major comment 1: In Fig 5g, the authors shows ChIP-seq data for H3K27ac, ARID1A and BRD4 in bone marrow cells treated with RANKL. To give a message that the colocalization of ARID1A with BRD4 is dependent on RANK signaling, please show the ChIPs before RANKL treatment as well. The promoter region of *Nfatc1* gene also should be pointed by an arrow in the genome browser tracks.

Major comment 2: In Fig 5h, the author shows de novo motifs for ARID1A or ARID1A-dependent BRD4 bindings. Please provide total peak numbers of their bindings and what is the background used for the motif analyses.

Major comment 3: Although the authors predict the colocalization of PU.1 with ARID1A in Figs 5i and 5j, PU.1 ChIP-seq data should be more reasonable to confirm it by adding the track in Fig 5g.

Major comment 4: To confirm if the promoter-enhancer proximity on *Nfatc1* gene is functional, the promoter luciferase activity assay should be performed after generating mini genes with the

promoter-enhancer connection. If it is functional for Nfatc1 promoter activity, 3C-qPCR should be performed to evaluate if the promoter-enhancer proximity is induced during osteoclastogenesis. Overall, the authors can illustrate the model as shown in Fig 5n.

Major comment 5: The authors used Nfatc1 lentivirus and Arid1a siRNA in osteoclast culture as shown in Fig 4 and Fig 6, respectively. Please describe when the lentivirus or siRNA was introduced in cells and how long the cells were cultured with or without the lentivirus or siRNA until the cell harvest. The method "Lentivirus and siRNA transfection" needs more detailed information. The final concentration of siRNA (25 mM) looks very high. How did the authors optimize the concentration? Did the authors also evaluate some experiments by using another Nfatc1 siRNA?

Major comment 6: The expression of STAT1 is described in lines 351-357. To remind of why the authors focused on STAT1/IFN γ signaling here, please describe the reason along with the reference #18 before showing Figs 6g-j.

Major comment 7: BRD9 heterozygous knockout mice were used in Figs 7h-l. Why were not the homozygous mice utilized?

Major comment 8: In Extended Data Fig 7d, BRD9 protein expression is shown in a SP141 inhibitor dose-dependent manner, but the BRD9 expression levels are similar between basal condition and 0.25 μ M SP141. If the authors cannot explain why it happened, they might not have to show the dosage to avoid confusion.

Major comment 9: ARID1A-deficiency suppresses osteoclastogenesis (CTSK expression) as well as the treatment of BRD4 inhibitor JQ1 in Fig 8a. On the other hand, STAT1 expression in ARID1A KO cells is same as in the control conditions in Figs 8a and e. Please describe the mechanisms clearly by illustrating the distinguished protein complex models.

Major comment 10: The author showed c-Fos expression in Fig 8f. It has been known that the AP-1 family transcription factor is induced by RANKL treatment. However, it seems that showing c-Fos expression is not essential and actually there is not any description in this manuscript. Therefore, the c-Fos blot should be replaced by NFATc1 blot.

Miner comments

Major comment 1: In lines 73-76, the authors gave examples of ARID1A function. ARID1A has been shown previously to have roles to play in thermogenesis accompanied by the promoter-enhancer

proximity of thermogenic genes depending on beta-adrenergic signaling (PMID: 25948511). Please describe the chromatin remodeler function of ARID1A by referring to this publication, which should support the mechanism in Fig 5.

Miner comment 2: Typically, BMDM means bone marrow derived macrophages. To avoid confusion, bone marrow cell (BMC) should be appropriate when undifferentiated bone marrow cell is indicated.

Miner comment 3: Please describe the details about Fig 1c in the result section.

Miner comment 4: Which sex was used for in vivo bone analyses (Figs 1 and 7)? Is there any difference between male and female?

Miner comment 5: Please include Mmp9 mRNA expression in Figs 4f and 7f to compare to other data sets (e.g., Figs 5b and 6c).

Miner comment 6: The letters in some figures are so small and it is difficult to read. Especially, Figs 3d-g (x and y-axis), Figs 4d-e, Fig 7b (x and y-axis), Fig 8c (x and y-axis), Extended Data Figs 3a-e, 4a-h and 7b should be edited clearly.

Miner comment 7: In Fig 3c, “Osteoclast” and “Neutrophils” do not point out the exact numbers.

Miner comment 8: Molecular weight size markers should be indicated in each picture for western blots (Figs 2h, 4l, 5k, 6a, 6d, 6g, 6i, 7g, 8a, 8d, 8e, and 8f, Extended Data Figs 1c, 5b, 5d, and 7e). In addition, all uncropped images of western blots should be shown as Extended data.

Miner comment 9: The positions of primers used in PU.1 ChIP-qPCR should be indicated in Fig 5g.

Reviewer #3 (Remarks to the Author):

The manuscript by Du et al identifies a new role for the chromatin remodeler ARID1A in osteoclast fate commitment. The authors provide evidence that ARID1A expression is induced during osteoclastogenesis and that its conditional deletion in myeloid/granulocyte lineages (i.e. Lyz2-Cre)

corresponds with elevated bone mass in mice associated with reduced osteoclast numbers. By combining single cell sequencing with bioinformatics, and CHIP-seq and the authors identified a number of enhancer regions occupied by ARID1A. Moreover, they show that loss of ARID1A leads to a down regulation of the osteoclast master transcription factor Nfatc1. In addition, they provide evidence to suggest that ARID1A activates Nfatc1 via the formation of nuclear condensates in collaboration with coactivator BRD4 and transcriptional factor PU.1 within Nfatc1 super enhancer sites. The authors further demonstrate competitive antagonism between the ARID1A and BRD9 via BRD4 thereby unveiling a new mechanism to control and safeguard cell commitment and fate during osteoclastogenesis.

Conceptually this is a very interesting paper, and the bioinformatic and mechanistic data, for the most part, is of high quality and appropriately controlled. However, in their current state, the physiological vivo data remains descriptive and lacks quantitation needed to support the conclusions drawn. Specifically, the bone and cellular phenotyping of ARID1A KO mice remain too preliminary to conclude that ARID1A plays a crucial/indispensable role in osteoclastogenesis in vivo and therefore these data should be improved.

Major comments:

- The bone phenotype data presented in Figures 1, 2 and 7 remain underdeveloped and need to be improved to support the authors claims that ARID1A plays an “indispensable” role in osteoclastogenesis. Key parameters are missing, most notably the sex-specific comparisons between male and female mice and cortical bone parameters etc. The number of animals used (i.e. n=3) is also insufficient to support statistical robustness, especially considering the modest bone phenotype observed upon ARID1A KO so additional numbers are required to bolster these data. Is there any change observed on the cortical bone parameters, thickness, endosteal/periosteal surfaces? These data should be included.
- Much of the data presented in Figures 2 and 7j-k are limited to the epiphysis of very young mice i.e. 3-6 weeks of age at which time they remain skeletally immature. Therefore, it would be important to show that the bone phenotype is maintained at later age points in skeletally mature animals e.g. >3-6 months as bone phenotypes can often resolve with age.
- It is too premature to conclude that reduced osteoclast number and thus reduced bone resorption is the major contributor to the bone phenotype. Line 150 states “we individually examined osteoclastic bone resorption”, however these parameters were not assessed/presented. To measure resorption, eroded surface parameters are required by histomorphometry. Additionally, osteopetrotic phenotypes are typically associated with persistence of unresorbed calcified cartilage. Can the authors provide evidence of this? This would go some way to supporting claims that the bone phenotype is osteoclast driven.
- Likewise to assess bone formation, osteoid surfaces and bone-lining osteoblast numbers need to be assessed. The data presented in Figure 2a-e lack quantification and are therefore insufficient to support claims that osteoblasts do not influence the bone phenotype. This is important because LysM targets multiple myeloid populations, some of which that are known to directly modulate the recruitment of osteoprogenitors cells to bone surfaces. Bone formation and mineralization rates therefore need to be assessed. The single Day 7-calcein injection described in the Methods (lines

512-13) and depicted in Figure 1e is not sufficient to measure bone formation/mineralization and is therefore misleading to the reader. Dual fluorophore labelling is required to assess mineral apposition and bone formation rates dynamically. Similarly, Von Kossa staining of mineralization should be performed on undecalcified samples, not paraffin embedded decalcified sections as detailed in lines 505-509 of the Methods.

- The histological data should be supported by biochemical markers of bone turnover i.e. TRAP and CTX1 for osteoclasts, OCN and PN1P for osteoblasts to strengthen the resolve of the conclusions drawn, as is standard for the field.
- Figure 3- The experimental plan in Figure 3A indicates that cells were subjected to a 24 RANKL stimulation which although can elicit expression of pro-osteoclast genes (Mmp9, ACp5 and Dcstamp) is not sufficient to drive the formation of “terminal differentiated osteoclasts” as claimed for “cell cluster 3” in Figs 3B,C,E, F etc. It would be more appropriate to refer to Cluster 3 as “OC precursors” throughout the manuscript and to tone down any claim that this cluster is representative of mature osteoclasts in single-cell seq data and throughout the manuscript.
- Figure 3 panels h and I require quantification to support the conclusions.

Minor.

- Figure 1. Can the authors clarify why the immunofluorescence of ARID1A in Figure 1a is widely localized outside of the Tdtomato and CTSK + cells. If the staining of ARID1A and CTSK is specific then I would expect a much higher degree of colocalization? Can you provide evidence for the specificity of the antibodies for immunostaining. It is important to present quantification of the % of tomato/CTSK+ve cells that are ARID1A positive to support the statement on line 126-7 that “ARID1A is highly expressed in CTSK+ OCS on the surface of trabecular bone...”
- All immunoblots in figures lack molecular weights, these need to be included.
- Please detail how was the dose of JQ1 determined? Was an IC50 established?
- Figure 7i, please included the BV/TV values here as it is a standard parameter. Animal numbers also need to be increased to strengthen the resolve of the phenotype presented.
- LysM and Lyz2 -Cre are used interchangeably between the text and figures (e.g. see Figure 2a-f compared with Fig 2h). Please be consistent throughout the manuscript.
- Discussion of the implications of these findings in the context of bone homeostasis and potential therapeutic application for bone disease would be a nice addition to the Discussion section.
- Pg 5 line 88....in cell fate decisions is “unveiled” should read “unknown/unclear”?

May 1, 2024

RE: Nature Communications manuscript NCOMMS-23-52629-T

Response to Reviewers' comments:

Reviewer #1:

In this study, the authors showed that chromatin remodeler ARID1A critically contributes to the osteoclast fate determination. Using genetic loss-of-function experiments, genome-wide epigenetic analyses and single-cell RNAseq, the authors demonstrated that ARID1A activates NFATc1 expression by binding to the super enhancer region with BRD4 and PU.1. The experiments are well controlled, results are clear and the conclusion appears sound, however, there are several important points to be addressed.

1) The molecular mechanisms underlying NFATc1 induction have been well studied in osteoclast research field. In this study, the authors showed that the ARID1A/BRD4/PU.1 condensate is a critical determinant of NFATc1 induction but previous studies have shown that NF-kB, AP-1 and NFATc1 itself contribute to the NFATc1 induction by activating cis-regulatory elements. Do these factors physically associate with the ARID1A/BRD4/PU.1 condensate and regulate the same super enhancer region? The authors should clarify the relationship between the ARID1A/BRD4/PU.1 complex and other previously reported factors (NF-kB, AP-1 and NFATc1) in the regulation of NFATc1 induction.

Thank the reviewer for these comments. We found the NF-kB, AP-1 and NFATc1 motifs are also enriched in the same *Nfatc1* SE region, which suggests NF-kB, AP-1 and NFATc1 itself could also participate in the ARID1A/BRD4/PU.1 condensate formation at *Nfatc1* SE region and facilitate its transcription activation during safeguarding the canalization of osteoclastogenesis. In particular, we validated the physical association between NFATc1 with ARID1A using co-immunoprecipitation assay during osteoclastogenesis (**Response letter Figure 1**). We listed the motif locations of NF-kB,

AP-1 and NFATc1 in the same *Nfatc1* SE region in (Supplementary Fig. 8a, b in revised manuscript) and clarified their potential relationship in the Result section.

Response letter Figure 1. Physical association between NFATc1 with ARID1A during osteoclastogenesis. Co-IP assay with ARID1A antibody (or IgG) in BMCs during osteoclastic induction, followed by immunoblotting of ARID1A and NFATc1 (with gradually increased exposure time).

2) This reviewer feels that the ARID1A story is sufficiently novel and interesting but the BRD9/STAT1 story makes this paper very complicated. The concluding schema in Figure.8 is confusing and does not reflect the whole story or main message of this manuscript. The authors should remove or revise the schema and please clearly discuss how ARID1 complex and BRD9 complex regulate each other.

Thank the reviewer for these comments. We removed the confusing schema and added description on potential regulatory mechanism between ARID1A complex and BRD9 complex in the Discussion section.

3) It is interesting that the authors' scRNAseq data suggested that ARID1a regulates the transition of proliferation-differentiation during osteoclastogenesis. Previous osteoclast scRNAseq papers showed Cited2 (Nat Metab. 12:1382-1390, 2020) and Rab38 (JBMR Plus . 16;6:e10631. 2022) also control the transition of proliferation-differentiation in osteoclasts. The authors should examine the expression levels of Cited2 and Rab38 in Arid1a KO osteoclasts in Fig.2g and discuss the possibility that ARID1a is involved in

the cell-cycle regulation during osteoclastogenesis.

Thank the reviewer for these suggestions. We have examined the expression levels of *Cited2* and *Rab38* in ARID1A-depleted OCs (**Supplementary Fig. 5 in revised manuscript**). The results showed that loss of *Arid1a* could also lead to decreased expression of *Cited2* and *Rab38*, suggesting besides failed activation of *Nfatc1* validated in our study, the downregulated expression of other critical factors, such as *Cited2* and *Rab38* could also contribute to the defective transition of proliferation-differentiation in OCs after the loss of *Arid1a*. We added the new findings and cited these two references in the **Result section** of our revised manuscript.

Besides, we elucidated the possibility that ARID1A is involved in the cell-cycle regulation during osteoclastogenesis in the **Discussion section** accordingly.

4) Why IFN- β signaling is activated in ARID1a KO osteoclasts (Fig7b,c)? Does this mean IFN- β production is elevated and this acts on osteoclasts in an autocrine manner? The authors should show the expression levels of IFN- β in ARID1a KO osteoclasts.

Thank the reviewer for this suggestion. Our previous study demonstrated that the BRD9-ncBAF complex negatively regulates OC differentiation through STAT1/IFN- β signaling activation (Du J, et al. *Nat Commun.* 2023 Mar 14;14(1):1413.). In the present study, we detected that ARID1A loss leads to increased protein level of BRD9 (Fig. 7a) and IFN- β signaling (Fig. 7b and Supplementary Fig. 9) in BMCs from *LysM-Cre;Arid1a^{fl/fl}* mouse comparing with the control group after RANKL induction. Then we validated that the increased STAT1 in BMCs after loss of ARID1A was rescued in BRD9 heterozygous knockout in *LysM-Cre;Arid1a^{fl/fl}* mice (Fig. 8g). We have examined the expression levels of *Ifnb1* in *Arid1a* KO osteoclasts. The results showed that loss of ARID1A could lead to increased expression of *Ifnb1*. The elevated IFN- β production could also act on osteoclasts in an autocrine manner, further contributing to the increased IFN- β signaling. We added this data (**Fig. 7c in revised manuscript**) and described it in the **Result section** accordingly.

5) Quantification of bone histomorphometry is needed in Figure2a-e and Figure7 j,k.

Thank the reviewer for this suggestion and we added the quantification data in our revised manuscript (**Fig. 2a-h** and **Fig. 8c-f**) accordingly.

6) The physical association of ARID1A/ BRD4/PU.1 should be confirmed by immunoprecipitation.

Thank the reviewer for this comment. We have highlighted the result in our revised manuscript (**Fig. 5k**).

7) There are several sentences where adverbs should be used instead of adjectives (e.g. page 5 line 90; BRD9/ncBAF negative regulates → negatively, page 9 line 174; ARID1A transcriptional promotes → transcriptionally, page 12 line 227; are high enriched → highly)

Thank the reviewer for this suggestion. We carefully corrected these sentences throughout the entire manuscript.

Reviewer #2:

In this manuscript, Du et al show that ARID1A complex accelerates osteoclastogenesis, while BRD9 complex negatively regulates osteoclast differentiation. The authors focused on BRD4 as a molecule to determine cell fate canalization during osteoclastogenesis, which is colocalized with ARID1A and PU.1 on *Nfatc1* gene. The inhibition of BRD4 suppresses BRD9-KD induced osteoclast differentiation. The proposed pathway is interesting and unifies the previous finding (Du et al., 2023) about BRD9-mediated negative regulation of osteoclastogenesis. However, there are several major concerns about the putative mechanism. The biggest issue is that the proximal mechanism by which RANK signaling induces the enhancer-promoter proximity of *Nfatc1* gene that is dependent on ARID1A/BRD4/PU.1 complex formation is not clear.

Major comments

Major comment 1: In Fig 5g, the authors shows ChIP-seq data for H3K27ac, ARID1A and BRD4 in bone marrow cells treated with RANKL. To give a message that the colocalization of ARID1A with BRD4 is dependent on RANK signaling, please show

the ChIPs before RANKL treatment as well. The promoter region of *Nfatc1* gene also should be pointed by an arrow in the genome browser tracks.

Thank the reviewer for this suggestion. We added the ChIP-seq data for ARID1A and BRD4 in bone marrow cell before RANKL treatment (**Fig. 5g in revised manuscript**). There is no apparent peak of ARID1A and BRD4 in the *Nfatc1 SE* region, which suggested that the colocalization of ARID1A with BRD4 is dependent on RANK signaling. The promoter region¹ of the *Nfatc1* gene was pointed by an arrow (**Fig. 5g in revised manuscript**).

Major comment 2: In Fig 5h, the author shows de novo motifs for ARID1A or ARID1A-dependent BRD4 bindings. Please provide total peak numbers of their bindings and what is the background used for the motif analyses.

Thank the reviewer for this comment. Our ChIP-Seq data showed there are 828 peaks for ARID1A ($p < 0.001$), and 1298 peaks for ARID1A-dependent BRD4 binding ($p < 0.001$). For motif enrichment analysis we used AME with the same background model and negative control. AME tool of the MEME suite provides motifs that are enriched compared to the shuffled background (McLeay, R.C. & Bailey, T.L. Motif Enrichment Analysis: a unified framework and an evaluation on ChIP data. BMC Bioinformatics 11, 165 (2010).). We set the E-value threshold ($E\text{-value} \leq 10$, Fisher's exact test) for reporting enriched motifs. We used motif database JASPAR2022_CORE_vertbrates_non-redundant_v2. meme to test for enrichment. We added these descriptions in the **Result section** and analysis details in the **Methods section** of our revised manuscript.

Major comment 3: Although the authors predict the colocalization of PU.1 with ARID1A in Figs 5i and 5j, PU.1 ChIP-seq data should be more reasonable to confirm it by adding the track in Fig 5g.

Thank the reviewer for this comment. We further conducted PU.1 ChIP-seq and found the binding profile of PU.1 is highly similar with ARID1A and BRD4 in the *Nfatc1 SE* region. We have replaced the prediction data with PU.1 ChIP-seq data (**Fig. 5i in our**

revised manuscript).

Major comment 4: To confirm if the promoter-enhancer proximity on *Nfatc1* gene is functional, the promoter luciferase activity assay should be performed after generating mini genes with the promoter-enhancer connection. If it is functional for *Nfatc1* promoter activity, 3C-qPCR should be performed to evaluate if the promoter-enhancer proximity is induced during osteoclastogenesis. Overall, the authors can illustrate the model as shown in Fig 5n.

Thank the reviewer for this valuable suggestion. To further confirm if these enhancers on the *Nfatc1* gene are functional, the promoter luciferase activity assay was performed after generating mini genes with the E-P connection. Based on our ChIP-seq data of ARID1A and H3K27Ac in BMCs during osteoclastogenesis (**Fig. 4i in our revised manuscript**), we annotated their binding peaks as presentative enhancer regions at upstream (*Enhancer 1, E1*) and downstream (*Enhancer 2 and 3, E2 and E3*) of the *Nfatc1* promoter¹. The results showed that *E1, E2, and E3* together exhibit a much stronger activation function for *Nfatc1* promoter activity (**Fig. 4i in our revised manuscript**).

The remote enhancers *E2* and *E3* marked by H3K27ac located 10 and 21 kb downstream of the *Nfatc1* promoter, which are ideal distances to examine the long-range looping model by 3C analysis. Therefore 3C-qPCR was performed to evaluate the E-P proximity on the *Nfatc1* gene during osteoclastogenesis. Using the *Nfatc1* promoter as an anchor, we identified there is loop formation between *E2, E3* with the *Nfatc1* promoter region during osteoclastogenesis (**Fig. 4j in our revised manuscript**). While ARID1A depletion could reduce the long-range looping signal in response to RANKL-induction (**Fig. 4k in our revised manuscript**), suggesting the ARID1A-mediated transcriptional activation of *Nfatc1* could be partially through facilitating RANKL-induced E-P interactions.

We added these new evidences in the **Result section** of our revised manuscript.

Major comment 5: The authors used *Nfatc1* lentivirus and *Arid1a* siRNA in osteoclast

culture as shown in Fig 4 and Fig 6, respectively. Please describe when the lentivirus or siRNA was introduced in cells and how long the cells were cultured with or without the lentivirus or siRNA until the cell harvest. The method “Lentivirus and siRNA transfection” needs more detailed information. The final concentration of siRNA (25 mM) looks very high. How did the authors optimize the concentration? Did the authors also evaluate some experiments by using another *Nfatc1* siRNA?

Thank the reviewer a lot for this comment, we added detailed information in the method “Lentivirus and siRNA transfection” and also corrected the concentration (The final concentration of siRNA is 25 nM).

We have evaluated the function of *Nfatc1* siRNA. It shows that knockdown *Nfatc1* using siRNA in BMCs from *LysM-Cre;BRD9^{fl/fl}* mouse could also partially rescue the enhanced OC differentiation process with inhibited expression of MMP9 and CTSK (*Response letter Figure 2*).

Response letter Figure 2. *Nfatc1* siRNA inhibited excessive expression of MMP9 and CTSK in BMCs from *LysM-Cre;BRD9^{fl/fl}* mouse compared with control group after RANKL-induction, as measured by western blot.

Major comment 6: The expression of STAT1 is described in lines 351-357. To remind of why the authors focused on STAT1/IFNβ signaling here, please describe the reason along with the reference #18 before showing Figs 6g-j.

Thank the reviewer for this suggestion. We carefully revised the manuscript accordingly.

Major comment 7: BRD9 heterozygous knockout mice were used in Figs 7h-l. Why were not the homozygous mice utilized?

We have evaluated the bone phenotype of the femurs from both *LysM-Cre;Arid1a^{fl/fl};Brd9^{fl/+}* and *LysM-Cre;Arid1a^{fl/fl};Brd9^{fl/fl}* male mice at 3-month-old. The result showed there is a similar bone mass between these two groups, based on the X-ray examination (**Response letter Figure 3**).

Response letter Figure 3. X-ray analysis of the femurs from male mice with different genotypes at 3-month-old. Scale bar, 2 mm.

Major comment 8: In Extended Data Fig 7d, BRD9 protein expression is shown in a SP141 inhibitor dose-dependent manner, but the BRD9 expression levels are similar between basal condition and 0.25 μ M SP141. If the authors cannot explain why it happened, they might not have to show the dosage to avoid confusion.

Thank the reviewer for this suggestion. The exact dose-dependent function of SP141 needs to be further investigated and we revised the result to avoid confusion (**Supplementary Fig. 10e in our revised manuscript**).

Major comment 9: ARID1A-deficiency suppresses osteoclastogenesis (CTSK expression) as well as the treatment of BRD4 inhibitor JQ1 in Fig 8a. On the other hand, STAT1 expression in ARID1A KO cells is same as in the control conditions in Figs 8a and e. Please describe the mechanisms clearly by illustrating the distinguished protein complex models.

Thank the reviewer for this suggestion. We described the mechanisms clearly by illustrating the distinguished protein complex models accordingly in the **Result** and **Discussion section** of our revised manuscript.

Major comment 10: The author showed c-Fos expression in Fig 8f. It has been known that the AP-1 family transcription factor is induced by RANKL treatment. However, it seems that showing c-Fos expression is not essential and actually there is not any description in this manuscript. Therefore, the c-Fos blot should be replaced by NFATc1 blot.

Thank the reviewer for this suggestion. We replaced the c-Fos blot by NFATc1 blot (**Fig. 9f in our revised manuscript**).

Minor comments

Major comment 1: In lines 73-76, the authors gave examples of ARID1A function. ARID1A has been shown previously to have roles to play in thermogenesis accompanied by the promoter-enhancer proximity of thermogenic genes depending on beta-adrenergic signaling (PMID: 25948511). Please describe the chromatin remodeler function of ARID1A by referring to this publication, which should support the mechanism in Fig 5.

Thank the reviewer for this suggestion. We carefully learned this publication and found it much reference value for elevating our study. We cited this publication accordingly in the **Introduction** and **Result section** of our revised manuscript.

Minor comment 2: Typically, BMDM means bone marrow derived macrophages. To avoid confusion, bone marrow cell (BMC) should be appropriate when undifferentiated bone marrow cell is indicated.

Thank the reviewer for this comment, we have carefully revised this term accordingly throughout the entire manuscript.

Minor comment 3: Please describe the details about Fig 1c in the result section.

We added a detailed description of Fig 1c in the **Result section** of our revised

manuscript accordingly.

Miner comment 4: Which sex was used for in vivo bone analyses (Figs 1 and 7)? Is there any difference between male and female?

The phenotype after loss of *Arid1a* is consistent in both males and females. We added detailed gender information in the **Result section** and the **Figure legends** of our revised manuscript accordingly.

Miner comment 5: Please include Mmp9 mRNA expression in Figs 4f and 7f to compare to other data sets (e.g., Figs 5b and 6c).

We have added Mmp9 mRNA expression accordingly (**Supplementary Fig. 4 and Fig. 7e in our revised manuscript**).

Miner comment 6: The letters in some figures are so small and it is difficult to read. Especially, Figs 3d-g (x and y-axis), Figs 4d-e, Fig 7b (x and y-axis), Fig 8c (x and y-axis), Extended Data Figs 3a-e, 4a-h (CHIP SEQ) and 7b should be edited clearly.

Thank the reviewer for this comment. We re-edited the inappropriate letters accordingly throughout the entire manuscript.

Miner comment 7: In Fig 3c, “Osteoclast” and “Neutrophils” do not point out the exact numbers.

There are 1055 osteoclasts (18.15%) and 18 neutrophils (0.31%) in control group and 403 osteoclasts (7.6%) and 79 neutrophils (1.49%) in *LysM-Cre;Arid1a^{fl/fl}* group. We added the exact numbers in each cluster (**Supplementary Fig. 3a in our revised manuscript**).

Miner comment 8: Molecular weight size markers should be indicated in each picture for western blots (Figs 2h, 4l, 5k, 6a, 6d, 6g, 6i, 7g, 8a, 8d, 8e, and 8f, Extended Data Figs 1c, 5b, 5d, and 7e). In addition, all uncropped images of western blots should be shown as Extended data.

Thank the reviewer for this reminder. We added molecular weight size markers in each

picture for western blots throughout the entire manuscript. And all uncropped images of western blots have been provided in the **Source data** with the revised manuscript.

Miner comment 9: The positions of primers used in PU.1 ChIP-qPCR should be indicated in Fig 5g.

Thank the reviewer for this suggestion. We indicated the positions of primers used in PU.1 ChIP-qPCR accordingly (**Fig 5i in our revised manuscript**).

Reviewer #3:

The manuscript by Du et al identifies a new role for the chromatin remodeler ARID1A in osteoclast fate commitment. The authors provide evidence that ARID1A expression is induced during osteoclastogenesis and that its conditional deletion in myeloid/granulocyte lineages (i.e. Lyz2-Cre) corresponds with elevated bone mass in mice associated with reduced osteoclast numbers. By combining single cell sequencing with bioinformatics, and CHIP-seq and the authors identified a number of enhancer regions occupied by ARID1A. Moreover, they show that loss of ARID1A leads to a down regulation of the osteoclast master transcription factor Nfatc1. In addition, they provide evidence to suggest that ARID1A activates Nfatc1 via the formation of nuclear condensates in collaboration with coactivator BRD4 and transcriptional factor PU.1 within Nfatc1 super enhancer sites. The authors further demonstrate competitive antagonism between the ARID1A and BRD9 via BRD4 thereby unveiling a new mechanism to control and safeguard cell commitment and fate during osteoclastogenesis. Conceptually this is a very interesting paper, and the bioinformatic and mechanistic data, for the most part, is of high quality and appropriately controlled. However, in their current state, the physiological vivo data remains descriptive and lacks quantitation needed to support the conclusions drawn. Specifically, the bone and cellular phenotyping of ARID1A KO mice remain too preliminary to conclude that ARID1A plays a crucial/indispensable role in osteoclastogenesis in vivo and therefore these data should be improved.

Major comments:

- The bone phenotype data presented in Figures 1, 2 and 7 remain underdeveloped and need to be improved to support the authors claims that ARID1A plays an “indispensable” role in osteoclastogenesis. Key parameters are missing, most notably the sex-specific comparisons between male and female mice and cortical bone parameters etc. The number of animals used (i.e. n=3) is also insufficient to support statistical robustness, especially considering the modest bone phenotype observed upon ARID1A KO so additional numbers are required to bolster these data. Is there any change observed on the cortical bone parameters, thickness, endosteal/periosteal surfaces? These data should be included. Figure 7i, please included the BV/TV values here as it is a standard parameter. Animal numbers also need to be increased to strengthen the resolve of the phenotype presented.

Thank the reviewer for these suggestions. We increased the number of animals used (i.e. $n = 7$ for **Fig 1g**, and $n = 5$ for **Fig 8b in our revised manuscript**) in the related analysis. The phenotype of the trabecular bone is consistent between male and female mice (3-month-old male in **Fig. 1f, g**; 3-month-old female in **Fig. 8a, b in our revised manuscript**).

There is no significant change in the cortical bone parameters between control and ARID1A KO mice based on the CT quantification data, such as mean total crosssectional bone area (B. Ar) and mean total crosssectional bone perimeter (B. Pm) in neither males (3-month-old male in **Fig. 1f, g in our revised manuscript**) or females (6-month-old female in **Supplemental Fig. 2a, b in our revised manuscript**). We speculated that there could be some other redundant regulatory mechanism for osteoclastogenesis after loss of *Arid1a* in cortical bone. Consistently, it has been reported that different properties of bone remodeling or modeling activity exist between trabecular and cortical bone²⁻⁴.

We included the BV/TV values in **Fig. 1g**, **Fig. 8b**, and **Supplemental Fig. 2b** in our revised manuscript.

We added all above data and the description in the **Result section** accordingly.

- Much of the data presented in Figures 2 and 7j-k (HE CTSK) are limited to the epiphysis of very young mice i.e. 3-6 weeks of age at which time they remain skeletally immature. Therefore, it would be important to show that the bone phenotype is maintained at later age points in skeletally mature animals e.g. >3-6 months as bone phenotypes can often resolve with age.

Thank the reviewer for this advice. To validate the bone phenotype is maintained at later age points in skeletally mature animals, we added the data of representative micro-CT image and quantification analysis (**Supplemental Fig. 2a, b**), H&E staining (**Supplemental Fig. 2c, d**), and CTSK immunofluorescence (**Supplemental Fig. 2f, g**) of femurs from control and mutant mice at 6 months old, and substituted the data of representative micro-CT image and quantification analysis, H&E staining and CTSK immunofluorescence of femurs from 3-week-old (Figure 7j-k in previous manuscript) *LysM-Cre;Arid1a^{fl/fl}* mice, *LysM-Cre;Arid1a^{fl/fl};BRD9^{fl/+}* and littermate control mice with 3-month-old female mice in our revised manuscript (**Fig. 8a-f in our revised manuscript**).

- It is too premature to conclude that reduced osteoclast number and thus reduced bone resorption is the major contributor to the bone phenotype. Line 150 states “we individually examined osteoclastic bone resorption”, however these parameters were not assessed/presented. To measure resorption, eroded surface parameters are required by histomorphometry. Additionally, osteopetrotic phenotypes are typically associated with persistence of unresorbed calcified cartilage. Can the authors provide evidence of this? This would go some way to supporting claims that the bone phenotype is osteoclast driven.

Thank the reviewer for this suggestion and we added the quantification data of osteoclast surface per bone surface (Oc.S/BS) accordingly (**Fig. 2b in our revised manuscript**). We also added the Safranin O/fast green staining (**Fig. 2g, h in our revised manuscript**), revealing that the large persistence of unresorbed calcified cartilage underneath the growth plate of the distal femurs from *LysM-Cre;Arid1a^{fl/fl}* mice at 3 weeks of age. This result further suggests that osteopetrotic phenotypes caused

by ARID1A inactivation are typically associated with the persistence of unresorbed calcified cartilage. We added this new data accordingly and description in the **Result section** of our revised manuscript.

- Likewise to assess bone formation, osteoid surfaces and bone-lining osteoblast numbers need to be assessed. The data presented in Figure 2a-e lack quantification and are therefore insufficient to support claims that osteoblasts do not influence the bone phenotype. This is important because LysM targets multiple myeloid populations, some of which that are known to directly modulate the recruitment of osteoprogenitors cells to bone surfaces. Bone formation and mineralization rates therefore need to be assessed. The single Day 7-calcein injection described in the Methods (lines 512-13) and depicted in Figure 1e is not sufficient to measure bone formation/mineralization and is therefore misleading to the reader. Dual fluorophore labelling is required to assess mineral apposition and bone formation rates dynamically. Similarly, Von Kossa staining of mineralization should be performed on undecalcified samples, not paraffin embedded decalcified sections as detailed in lines 505-509 of the Methods.

Thank the reviewer for these valuable suggestions. Toluidine blue staining and osteoblast surface per bone surface (Ob.S/BS) was added accordingly (**Fig. 2c, d in our revised manuscript**).

To show osteoid that distinguished from mineralized bone, we conducted GRB staining⁵ and added the quantification data of osteoid surface per bone surface of trabecular bone (OS/BS) accordingly (**Fig. 2e, f in our revised manuscript**).

We also conducted the dual fluorophore labelling, there is no significant change on the BFR and MAR after loss of *Arid1a* (**Response letter Figure 4**).

Besides, Von Kossa staining of mineralization is indeed performed on undecalcified samples, so that we corrected the description in the **Methods section** accordingly in our revised manuscript.

Response letter Figure 4. Evaluation of dynamic bone formation using dual fluorophore labelling. Visualization and quantification of calcein/ARS labeling at 7 days and 2 days before collection at the distal femur in *LysM-Cre;Arid1a^{fl/fl}* mice and littermate control mice at the age of 12 weeks. $n = 6$. Data in this figure are represented as mean \pm SD. Two-tailed Student's *t*-test. Scale bar, 200 μm .

- The histological data should be supported by biochemical markers of bone turnover i.e. TRAP and CTX1 for osteoclasts, OCN and PN1P for osteoblasts to strengthen the resolve of the conclusions drawn, as is standard for the field.

Thank the reviewer for this suggestion. We conducted quantitative measurements of serum biomarkers of bone resorption (TRAP and CTX-I) and bone formation (PINP and OCN) in control and *LysM-Cre;Arid1a^{fl/fl}* male mice at 3-month-old. The result showed that loss of *Arid1a* leads to suppressed bone resorption, without apparent change on the bone formation ability. We added these data accordingly (Supplementary Fig. 2e in our revised manuscript).

- Figure 3- The experimental plan in Figure 3A indicates that cells were subjected to a 24 RANKL stimulation which although can elicit expression of pro-osteoclast genes (Mmp9, ACp5 and Dcstamp) is not sufficient to drive the formation of “terminal differentiated osteoclasts” as claimed for “cell cluster 3” in Figs 3B,C,E, F etc. It would be more appropriate to refer to Cluster 3 as “OC precursors” throughout the manuscript and to tone down any claim that this cluster is representative of mature osteoclasts in single-cell seq data and throughout the manuscript.

Thank the reviewer for this suggestion and we agree with it. We referred to Cluster 3 as “OC precursors” throughout the revised manuscript and toned down any claim that this cluster is representative of mature osteoclasts in single-cell seq data and throughout the

revised manuscript.

- Figure 3 panels h and I require quantification to support the conclusions.

Thank the reviewer for this suggestion and we added the quantification data in **Fig. 3h, i in our revised manuscript** accordingly.

Minor.

- Figure 1. Can the authors clarify why the immunofluorescence of ARID1A in Figure 1a is widely localized outside of the Tdtomato and CTSK + cells. If the staining of ARID1A and CTSK is specific then I would expect a much higher degree of colocalization? Can you provide evidence for the specificity of the antibodies for immunostaining. It is important to present quantification of the % of tomato/CTSK+ve cells that are ARID1A positive to support the statement on line 126-7 that “ARID1A is highly expressed in CTSK+ OCS on the surface of trabecular bone....”

Thank the reviewer for this comment. Although we find the expression level of ARID1A at mRNA and protein is increased after RANKL-induction during osteoclastogenesis, it is indeed that ARID1A expresses broadly. We validated the specificity of ARID1A antibody for immunostaining in the sections from *LysM-Cre;tdT* and *LysM-Cre;Arid1a^{fl/fl};tdT* mice. We have validated that ARID1A is KO efficiently in tdT+ myeloid lineage cells in *LysM-Cre;Arid1a^{fl/fl};tdT* mice, while it is unaffected in TdT- cell (**Supplementary Fig. 1c in our revised manuscript**). As a chromatin remodeler, it's the cofactor entitled ARID1A its functional specificity. This result is consistent with our previous publication⁶ (Du J, et al. Cell Rep. 2021 Apr 6;35(1):108964.). To make it clear, we revised the related descriptions for Fig. 1a in **the Result section** and added this new data accordingly in our revised manuscript.

- Please detail how was the dose of JQ1 determined? Was an IC50 established?

Thank the reviewer for this comment and the dose of JQ1 determined based on the IC50 evaluation. We found 0.08 μ M is the concentration of JQ1 that is required for 50% inhibition on the expression of MMP9 and CTSK (**Response letter Figure 5**)..

Response letter Figure 5. IC50 evaluation of JQ1 for inhibition on osteoclastogenesis. The protein expression of MMP9 and CTSK in JQ1 at different concentration or control vector treated BMCs from 4-week-old mice after RANKL-induction, as measured by western blot.

- All immunoblots in figures lack molecular weights, these need to be included.

Thank the reviewer for this reminder. We added molecular weight size markers in each picture for western blots in the revised manuscript.

- LysM and Lyz2 -Cre are used interchangeably between the text and figures (e.g. see Figure 2a-f compared with Fig 2h). Please be consistent throughout the manuscript.

Thank the reviewer for this reminder. We have revised it and made it consistent throughout the manuscript.

- Discussion of the implications of these findings in the context of bone homeostasis and potential therapeutic application for bone disease would be a nice addition to the Discussion section.

Thank the reviewer for this suggestion and we agree with it. We added the implications of these findings in the context of bone homeostasis and potential therapeutic application for bone disease in the **revised Discussion section**.

- Pg 5 line 88....in cell fate decisions is “unveiled” should read “unknown/unclear”?

Thank the reviewer for this reminder. We have revised it accordingly.

We greatly appreciate the editor and reviewers for these helpful comments, which have helped to improve our study. Thank you very much for your consideration.

Sincerely,

Xinquan Jiang, DDS, PhD

Dean of Faculty of Dentistry, Executive Dean of College of Stomatology, Shanghai Jiao Tong University

Deputy Director of National Center for Stomatology

Director of Shanghai Engineering Research Center of Advanced Dental Technology and Materials

E-mail: xinquanjiang@aliyun.com

Tel: +86-21-23271699

Reference

1. Chuvpilo, S. *et al.* Autoregulation of NFATc1/A expression facilitates effector T cells to escape from rapid apoptosis. *Immunity* **16**, 881-895 (2002).
2. Li, J. *et al.* Different bone remodeling levels of trabecular and cortical bone in response to changes in Wnt/beta-catenin signaling in mice. *J Orthop Res* **35**, 812-819 (2017).
3. Lerebours, C., Weinkamer, R., Roschger, A. & Buenzli, P.R. Mineral density differences between femoral cortical bone and trabecular bone are not explained by turnover rate alone. *Bone Rep* **13**, 100731 (2020).
4. Montagnani, A. Bone anabolics in osteoporosis: Actuality and perspectives. *World J Orthop* **5**, 247-254 (2014).
5. Gaytan, F., Morales, C., Reymundo, C. & Tena-Sempere, M. A novel RGB-trichrome staining method for routine histological analysis of musculoskeletal tissues. *Sci Rep* **10**, 16659 (2020).
6. Du, J. *et al.* Arid1a-Plagl1-Hh signaling is indispensable for differentiation-associated cell cycle arrest of tooth root progenitors. *Cell Rep* **35**, 108964 (2021).

REVIEWERS' COMMENTS

Reviewer #1 (Remarks to the Author):

Thank you for addressing all my remarks. I have no further comments.

Reviewer #2 (Remarks to the Author):

The revised manuscript has been improved with new supportive data and the reviewer is satisfied with the replies and additional experiments. However, this reviewer would like to request the authors to deal with some points before publication.

Comment 1: Fig. 5i shows the genome browser track for PU.1 ChIP-seq. It should be included in Fig. 5g along with ChIPs for ARID1A and BRD4.

Comment 2: Please describe the concentration and treatment time for reagents (e.g., formaldehyde, DpnII, T4 DNA Ligase) in the method section of 3C-quantitative polymerase chain reaction.

Comment 3: Supplemental Table S5 is missing.

Reviewer #3 (Remarks to the Author):

The authors have made considerable efforts to address my initial concerns. The additional data and quantitation have improved the overall quality of the manuscript and strengthened the resolve of the key conclusions drawn. I am satisfied that my major concerns have been adequately addressed.

Minor correction: Line 168-9 the defective “osteogenesis” should be corrected to “osteoclastogenesis”.

June 23, 2024

RE: Nature Communications manuscript NCOMMS-23-52629A

Response to Reviewers' comments:

Reviewer #1:

Thank you for addressing all my remarks. I have no further comments.

We feel great thanks for the reviewer's professional review work during the revision process.

Reviewer #2:

The revised manuscript has been improved with new supportive data and the reviewer is satisfied with the replies and additional experiments. However, this reviewer would like to request the authors to deal with some points before publication.

We thank the reviewer for all the constructive suggestions during the revision process.

Comment 1: Fig. 5i shows the genome browser track for PU.1 ChIP-seq. It should be included in Fig. 5g along with ChIPs for ARID1A and BRD4.

Thank the reviewer for this comment. We further included PU.1 ChIP-seq along with ChIPs for ARID1A and BRD4 in the same genome browser track (Fig. 5g in revised manuscript).

Comment 2: Please describe the concentration and treatment time for reagents (e.g., formaldehyde, DpnII, T4 DNA Ligase) in the method section of 3C-quantitative polymerase chain reaction.

Thank the reviewer for this suggestion. We added a detailed description in the method section of 3C-quantitative polymerase chain reaction in our revised manuscript accordingly.

Comment 3: Supplemental Table S5 is missing.

Thank the reviewer for this reminder. We have renamed "Supplementary Tables S1-5"

to "Supplementary Data 1-5" and provided each Excel sheet as a separate Excel file, i.e. "Supplementary Data 5", etc along with our revised manuscript.

Reviewer #3:

The authors have made considerable efforts to address my initial concerns. The additional data and quantitation have improved the overall quality of the manuscript and strengthened the resolve of the key conclusions drawn. I am satisfied that my major concerns have been adequately addressed.

Minor correction: Line 168-9 the defective "osteogenesis" should be corrected to "osteoclastogenesis".

Thank the reviewer for this reminder. We have revised it accordingly.

We greatly appreciate the editor and reviewers for these helpful comments, which have helped to improve our study a lot. Thank you very much for your consideration.

Sincerely,

Xinquan Jiang, DDS, PhD

Dean of Faculty of Dentistry, Executive Dean of College of Stomatology, Shanghai Jiao Tong University

Deputy Director of National Center for Stomatology

Director of Shanghai Engineering Research Center of Advanced Dental Technology and Materials

E-mail: xinquanjiang@aliyun.com

Tel: +86-21-23271699